# Explainable Visual Forgery Detection: A Survey

## Abstract

The rapid growth of AI-driven image manipulation technologies poses critical challenges for verifying content authenticity. While many forgery detection systems achieve high accuracy, their black-box nature limits deployment in high-stakes domains that demand transparency and explainability. This survey presents the first comprehensive review of explainable forgery detection in images and videos, introducing a novel taxonomy structured around three dimensions: Forgery Localization (FL), which pinpoints manipulated regions; Forgery Attribution (FA), which identifies manipulation sources; and Forgery Judgment Basis (FJB), which clarifies decision reasoning. We systematically analyze 48 state-of-the-art methods across single-modal and multi-modal settings, examining architectural innovations and explainability mechanisms. Four feature-driven strategies (RGB, frequency-domain, noise-texture, and representation learning) are reviewed in detail, highlighting their complementary strengths. Benchmark datasets and evaluation protocols are also compared, and open challenges are identified, including the need for standardized explanation formats, uncertainty quantification, and broader dataset coverage. By establishing this taxonomy and synthesizing recent progress, this survey lays a foundation for developing transparent and trustworthy forgery detection systems, supporting real-world applications in forensic analysis, news verification, and regulatory compliance.

## 1 Introduction

The rapid advancement of sophisticated image manipulation technologies has fundamentally reshaped the digital media landscape, posing unprecedented challenges to content authenticity and trust. In this survey, *visual forgery detection* refers to forgery detection in visual media, primarily images but also including video and image-centric multimodal content. From traditional editing techniques such as splicing and copy-move to modern AI-driven generative models including Generative Adversarial Networks (GANs) and Diffusion Models (DMs), the ability to create highly realistic forged content has become easily accessible to both experts and ordinary users. This evolution introduces concerning societal risks, as manipulated images are increasingly employed in misinformation campaigns, identity theft, fraud, and various forms of digital deception, with potential implications for journalism, legal proceedings, and public trust.

These escalating risks highlight the urgent need for more robust detection systems capable of addressing the growing sophistication of image forgeries. Recent progress in generative AI has significantly lowered the technical barrier to producing photorealistic synthetic content. Large-scale diffusion models such as STABLE DIFFUSION (Rombach et al., 2022a), DALL · E (Ramesh et al., 2021), and MIDJOURNEY (Midjourney, Inc., 2022) are capable of generating images that are almost indistinguishable from real photographs, making traditional detection techniques increasingly inadequate. Meanwhile, the emergence of multimodal manipulation, which simultaneously alters visual, textual, and audio content, has further complicated detection. In such cases, inconsistencies can be strategically distributed across modalities to evade analysis. As shown in Figure 1, the field of forgery detection has witnessed remarkable growth in recent years, with numerous methods proposed at top-tier venues. Early works focused primarily on binary classification systems that distinguish real from fake, while subsequent studies before 2024 concentrated on explanation through visual localization. Since 2024, the field has witnessed a shift toward deeper explainability, addressing tasks such as model attribution and manipulation analysis. Representative approaches such as TRUFOR (Guillaro et al.,

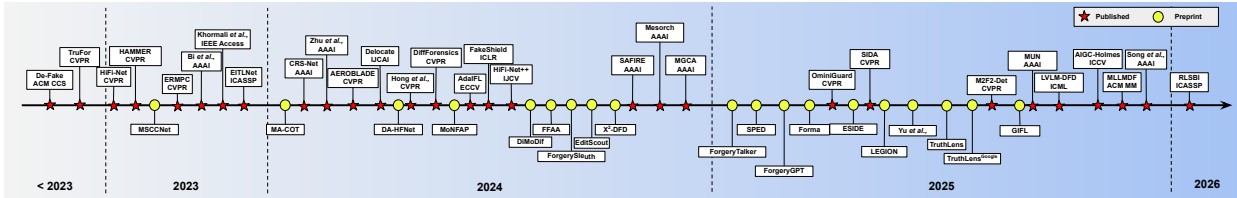

Figure 1: A roadmap of explainable visual forgery detection methods.

2023), HAMMER (Shao et al., 2024), FAKESHIELD (Xu et al., 2024), FORGERYSLEUTH (Sun et al., 2024), and TRUTHLENS (Kundu et al., 2025b) illustrate this progression toward more mature and explainable detection solutions.

Despite these advances, most existing forgery detection systems still operate as black-box models, providing little transparency into their decision-making processes. Although they may achieve high accuracy in controlled environments, their lack of explainability severely limits deployment in high-risk applications. In domains such as forensic analysis, news verification, regulatory compliance, and content moderation, understanding the reasoning behind detection outcomes is essential for building trust and accountability. This limitation becomes especially critical when the system produces erroneous predictions or when results require human verification. As detection systems are increasingly integrated into real-world applications, transparency, auditability, and human explainability are becoming essential requirements. Unlike conventional classifiers that provide only a binary label, explainable detection systems must deliver insights along three key dimensions: forgery localization (FL), which identifies the precise regions of manipulation; forgery attribution (FA), which determines the manipulation method, source, or origin; and forgery judgment basis (FJB), which clarifies the rationale behind the system's decision. Such multidimensional explainability not only enhances user trust and system reliability but also enables forensic analysts, journalists, and moderators to make well-informed decisions based on verifiable evidence.

Several surveys (Alam et al., 2021; Lin et al., 2024; Pei et al., 2024; Yu et al., 2024a; Xu et al., 2025) have reviewed image forgery detection and related areas, making significant contributions to understanding detection methodologies and architectural innovations. Table 1 shows that these surveys have effectively addressed technical performance, generalization capabilities, and specific application domains. However, the rapidly evolving landscape of AI-generated content and increasing deployment requirements in high-stakes applications have created new demands for systematic explainability analysis that extend beyond the scope of previous reviews. While *Zou et al.* (Zou et al., 2025) acknowledges the importance of explainability in AI-generated media detection, their work primarily focuses on detection techniques rather than establishing systematic taxonomies for multidimensional explainability analysis. As detection systems are increasingly deployed in critical applications requiring transparency and accountability, there emerges a clear opportunity for developing structured approaches to explainable detection that can address the fundamental questions of where, how, and why manipulations occur. More importantly, the scope of existing surveys is often constrained to specific application scenarios such as social media misinformation detection or facial deepfake detection, or to particular generative techniques such as Artificial Intelligence Generated Content (AIGC) or diffusion models. A comprehensive taxonomy centered on explainability, particularly one that spans multimodal detection capabilities, is still lacking.

To address this gap, this survey provides the first comprehensive review of explainable visual forgery detection. We introduce a taxonomy structured around three core explainable tasks: forgery localization (FL), forgery attribution (FA), and forgery judgment basis (FJB). This taxonomy goes beyond technical categorization by offering a structured understanding of how different approaches address spatial localization, source attribution, and decision reasoning. Building on this framework, we systematically analyze about 48 state-of-the-art methods across unimodal and multimodal paradigms, examining architectural innovations, feature extraction strategies, and mechanisms for explainability. Furthermore, we review four major feature-driven detection strategies, each offering distinct advantages in different manipulation scenarios: RGB features preserve spatial and color information and are effective for detecting boundary inconsistencies

Table 1: Comparison of existing related surveys on image and multimodal forgery detection in terms of architecture analysis, explainability, data modality, and scope. (**FL**: Forgery Localization; **FA**: Forgery Attribution; **FJB**: Forgery Judgement Basis.)

| Survey | Year | Architecture Analysis | Explainability | | | Data Modality | | Scope |
|--------|------|----------------------|----|----|-----|---------------|--------------|-------|
| | | | FL | FA | FJB | Single Modality | Multi Modality | |
| *Alam et al.,* (Alam et al., 2021) | 2021 | ✓ | ✗ | ✗ | ✗ | Image, Text, Video, Audio | - | Social Media Fake Image Detection |
| *Lin et al.,* (Lin et al., 2024) | 2024 | ✗ | ✗ | ✗ | ✗ | Images, Text, Video, Audio | Image-Text | AIGC Generated Detection |
| *Pei et al.,* (Pei et al., 2024) | 2024 | ✗ | ✗ | ✗ | ✗ | Image, Text, Audio | Image-Text, Image-Audio | Facial Deepfake Detection |
| *Yu et al.,* (Yu et al., 2024a) | 2024 | ✗ | ✗ | ✗ | ✗ | Image, Text, Video, Audio | Image-Text | AIGC Generated Detection |
| *Xu et al.,* (Xu et al., 2025) | 2025 | ✓ | ✗ | ✗ | ✗ | Image | Image-Text | Diffusion Generated Detection |
| *Zou et al.,* (Zou et al., 2025) | 2025 | ✗ | ✓ | ✗ | ✗ | Image, Text, Video, Audio | Image-Text, Image-Audio | AIGC Generated Detection |
| **Ours** | 2026 | ✓ | ✓ | ✓ | ✓ | Image, Video | Image-Text, Image-Audio | Explainable Visual Forgery Detection |

and occlusion mismatches; frequency-domain features capture periodic structures and spectral anomalies that are difficult to detect in the spatial domain; noise-texture features leverage high-pass filtering to capture statistical patterns of the imaging process, making them highly sensitive to subtle manipulations; and representation learning features automatically learn high-level semantic abstractions through deep networks, which are particularly important for multimodal detection. The main contributions of this survey can be summarized as follows:

- We propose a comprehensive taxonomy of forgery detection methods centered on three explainability dimensions: forgery localization (FL), forgery attribution (FA), and forgery judgment basis (FJB).

- We systematically analyze four feature-driven detection strategies, namely RGB features, frequency-domain features, noise-texture features, and representation learning features, and discuss their strengths across different manipulation scenarios.

- We provide an extensive review of unimodal and multimodal detection architectures, highlighting their design principles, explainability mechanisms, and implementation strategies.

- We offer a holistic overview of benchmark datasets and evaluation protocols that have been specifically developed for explainable detection tasks.

- We identify key challenges and outline promising directions for future research, aiming to guide the development of transparent, accountable, and trustworthy forgery detection systems.

For transparency and reproducibility, we provide a detailed description of our literature search strategy, including databases, search terms, time range, and inclusion/exclusion criteria, in Appendix 10.1. In the context of the ongoing arms race between generative and detection technologies, this survey aims to equip researchers, practitioners, and policymakers with a structured understanding of the current capabilities and limitations of explainable forgery detection. By establishing a clear taxonomy, highlighting technical innovations, and identifying future opportunities, we seek to accelerate progress toward detection systems that achieve not only high accuracy but also the transparency and explainability required for responsible deployment in trust-critical applications.

## 2 Related Work

The field of image forgery detection has experienced significant growth over the past decade, with multiple survey articles systematically examining this rapidly expanding research domain from various perspectives. *Alam et al.* (Alam et al., 2021) presented pioneering work focused on multimodal disinformation detection, with particular emphasis on content authenticity verification in social media environments. As shown in Table 1, their survey covered multiple modalities, including images, text, video, and audio, while providing architectural analysis primarily focused on social media fake image detection scenarios. Their research revealed correlations between factuality and harmfulness, finding that these correlations vary across different linguistic contexts, specifically, 56% of false Arabic content was also harmful, compared to 24% for English content. This work emphasized the critical role of multimodal content in disinformation propagation,

demonstrating that images and videos are more easily consumed, attract greater attention, are perceived as more credible, and spread further than plain text. *Lin et al.* (Lin et al., 2024) provided a comprehensive analysis of multimedia content detection generated by Large AI Models (LAIMs), introducing an innovative taxonomy categorized by media modality. Their survey covered text, images, videos, and audio generated by diffusion models and large language models, analyzing the field from two perspectives: "pure detection" (enhancing detection performance) and "beyond detection" (adding attributes like generalizability, robustness, and explainability). Their work encompassed images, text, video, audio, and image-text multimodal processing, establishing an important theoretical framework for the AIGC detection domain, though with limited development in specific explainability dimensions. *Pei et al.* (Pei et al., 2024) presented a specialized analysis of deepfake generation and detection, comprehensively covering tasks including face swapping, face reenactment, talking face generation, and facial attribute editing. Their survey provided a detailed technical analysis of diffusion-based methods, covering image, text, and audio processing while supporting image-text and image-audio multimodal detection, though primarily limited to facial-related deepfake detection scenarios. *Yu et al.* (Yu et al., 2024a) focused on the theoretical foundations and detection methods of Fake Artificial Intelligence Generated Content (FAIGC), proposing an innovative taxonomy encompassing three dimensions: FAIGC intent, modality generation technologies, and creation methods. This work systematically analyzed FAIGC detection techniques across images, text, video, and audio, supporting image-text multimodal processing and contributing valuable insights to FAIGC detection theory construction. *Xu et al.* (Xu et al., 2025) specialized in generalizable diffusion-generated image detection, demonstrating outstanding performance in architectural analysis and deeply exploring technical details and methodological innovations in diffusion model detection. Their survey covered image and image-text multimodal processing, providing important references for this cutting-edge technical domain, though with relatively limited explainability analysis. *Zou et al.* (Zou et al., 2025) provided a comprehensive perspective on the evolution from non-multimodal large language models to multimodal large language models, covering images, text, video, audio, and image-text and image-audio multimodal processing. This work offered valuable analysis in forgery localization, recognizing the importance of explainability in AI-generated media detection, though primarily focusing on detection techniques rather than systematic explainability analysis frameworks.

Although existing surveys have made valuable contributions within their specialized domains, these works lack systematic dimensional categorization from an explainability perspective. Most existing surveys focus on specific application scenarios or particular generative technologies, without providing a unified framework that comprehensively addresses the fundamental questions of `"where"`, `"how"`, and `"why"` in forgery detection. To address this critical need, this survey provides the first systematic analysis of visual forgery detection from an explainability perspective. We construct a taxonomy centered on three core explainable tasks: Forgery Localization (FL), Forgery Attribution (FA), and Forgery Judgment Basis (FJB), establishing a structured framework for understanding how different approaches handle spatial localization, source attribution, and decision reasoning. Compared to existing work, our survey is the first to provide comprehensive coverage from an explainability-centered perspective across the three dimensions of FL, FA, and FJB, while supporting both unimodal detection of images and videos, and multimodal detection capabilities for image-text and image-audio content.

## 3 Background

This section provides essential background on the evolution of content forgery techniques and the fundamental principles of forgery detection. We examine the progression from traditional manipulation methods to modern AI-driven approaches, and establish the mathematical foundations for detection systems. As generative models produce increasingly realistic synthetic content that challenges conventional detection methods, understanding both forgery mechanisms and detection formulations becomes crucial for developing more explainable frameworks.

### 3.1 Image Forgery Techniques

#### 3.1.1 Traditional Image Forgery Methods.

Before the rise of deep learning, traditional image forgeries relied on classical image processing and low-level signal manipulation. Despite their simplicity, these methods were capable of producing visually convincing results within the technological constraints of the time. The primary traditional visual forgery techniques can be broadly categorized into several types.

- Splicing (Ng & Chang, 2004) involves combining regions from different source images to manipulate the semantic content of the final image. This technique is frequently used in fabricated news or altered historical photographs (Verdoliva, 2020).

- Copy-Move (Fridrich et al., 2003) refers to duplicating a region within the same image, often to obscure certain elements or artificially increase visual complexity.

- Removal (Criminisi et al., 2004) entails deleting specific regions from an image, followed by inpainting or texture synthesis to plausibly fill the resulting gaps.

- Retouching (Chen et al., 2018) and Repainting (Criminisi et al., 2004) involve manually or semi-automatically redrawing selected areas, typically for restoration or object modification.

To enhance realism and reduce detection, traditional forgeries often incorporate auxiliary operations such as geometric transformations, including rotation, scaling, and perspective correction for spatial consistency; color and lighting adjustments for tonal uniformity; edge blending and gradient-domain fusion for seamless integration; and noise injection or compression simulation to mask traces of manipulation. These supporting techniques are typically layered on top of core forgery strategies to improve plausibility.

Despite these enhancements, such methods face notable limitations. They remain confined to the visual modality and often leave detectable artifacts in the frequency spectrum or compression residuals. Their dependence on manual intervention constrains scalability and automation. Most critically, they struggle to produce truly photorealistic results or operate at the scale required for mass content generation. These shortcomings, coupled with the rapid advances in deep learning, have propelled AI-driven forgery techniques to become the prevailing standard in digital content manipulation (Masood et al., 2023; Verdoliva, 2020).

#### 3.1.2 AI-Driven Image Forgery Methods.

The emergence of generative AI has greatly increased the complexity and subtlety of digital forgeries. Compared to traditional forgery techniques, AI-driven methods (Ramesh et al., 2022; Razzhigaev et al., 2023; CompVis, 2023) enable high-quality content generation with greater scalability and lower technical barriers. As deep learning advances, these forgery methods have achieved remarkable realism and precise control, posing significant challenges for detection systems.

**VAE-based Image Forgery Methods.** Early generative models, such as Variational Autoencoders (VAEs) (Kingma et al., 2013), established the foundation for AI-driven forgery by learning latent representations for image synthesis and editing. For example, $\beta$-VAEs (Higgins et al., 2017) enable detailed manipulation of facial attributes like age or gender, which can be used to obscure identity or create fraudulent documents and profiles. Vector Quantized Variational Autoencoders (VQ-VAEs) (Van Den Oord et al., 2017) generate high-quality facial reconstructions through discrete latent representations, making them effective tools for creating convincing face swaps and identity manipulations. Tools such as DeepFakes (dee, 2019) and FaceSwap (Kowalski, 2016) have lowered the technical threshold, enabling broader misuse.

**GAN-based Image Forgery Methods.** Generative Adversarial Networks (GANs) (Goodfellow et al., 2020) represent a major milestone in synthetic content creation, greatly enhancing the realism of generated images and enabling sophisticated forgery techniques. Early advances like ProGAN (Karras et al., 2017) and BigGAN (Brock et al., 2018) established high-resolution synthesis capabilities that could produce convincing fake images across diverse categories. StyleGANs (Karras et al., 2019) generate fully

synthetic, photorealistic images with unprecedented quality, creating entirely fabricated faces, objects, and scenes that are virtually indistinguishable from authentic content. Domain translation methods such as CYCLEGAN (Zhu et al., 2017) and STARGAN (Choi et al., 2018) enable sophisticated image manipulation by altering visual characteristics, including lighting, weather conditions, and artistic styles, to create misleading or deceptive content.

**DM-based Image Forgery Methods.** Diffusion models (DM) are currently at the forefront of generative forgery technologies, enabling the creation of highly realistic and structurally controllable synthetic content. Early breakthroughs such as DALL·E (Ramesh et al., 2022), IMAGEN (Saharia et al., 2022), and the open-source STABLE DIFFUSION (CompVis, 2023) demonstrated the capability of text-to-image generation to produce fabricated scenes and fake historical imagery directly from textual prompts. These models have significantly lowered the barrier for large-scale image forgery and accelerated the proliferation of synthetic content. In addition to general purpose synthesis, diffusion-based methods (Ruiz et al., 2023; Rombach et al., 2022b; Song et al., 2020b) have evolved to support more targeted and controlled generation. Personalization techniques like DREAMBOOTH (Ruiz et al., 2023) allow model fine-tuning with a handful of images, enabling the creation of highly specific forgeries involving particular individuals or branded objects. Beyond generation from scratch, modern diffusion-based inpainting techniques (Lugmayr et al., 2022; Zeng et al., 2021; Rombach et al., 2022b; CompVis, 2023) integrated into tools such as Adobe Firefly and RunwayML support seamless addition, removal, or alteration of image elements while maintaining visual realism. These tools blur the boundaries between original and manipulated content, making it increasingly difficult to distinguish authentic images from forged ones.

In a nutshell, these advancements not only enhance the visual fidelity of synthetic content but also introduce new challenges for forgery detection, especially when modifications are subtle, identity-targeted, or semantically aligned with the input prompt. The sophisticated control mechanisms in diffusion models enable highly localized edits that maintain global consistency, making traditional artifact-based detection methods less effective. Furthermore, the ability to generate contextually coherent content creates forgeries that are increasingly difficult to distinguish from authentic material through visual inspection alone.

### 3.1.3 Multi-Modal Forgery Methods.

While single-modal AI forgeries continue to advance rapidly, most real-world content combines multiple modalities such as text, images, and audio. Discrepancies between these modalities can reveal forgeries, but aligning them effectively increases the credibility of fake content. This challenge has led to the rise of multi-modal forgery techniques, which manipulate several modalities simultaneously to produce more convincing and difficult-to-detect disinformation.

**Image-Text Forgery.** In image-text forgery, diffusion models (CompVis, 2023; Ramesh et al., 2022; Rombach et al., 2022b) are commonly used to generate fabricated visuals aligned with false textual narratives, while large language models (LLMs), such as LAMA (Suvorov et al., 2022), CHATGPT (Achiam et al., 2023), GEMINI (Team et al., 2023), and VICUNA (Zheng et al., 2023), are employed to produce corresponding textual descriptions. For example, an attacker may simulate a traffic accident by generating coherent images and accompanying reports, thereby increasing the perceived authenticity. Incorporating multiple image perspectives further enhances the illusion of independent corroboration.

**Video-Audio Forgery.** In video-audio forgery, techniques such as WAV2LIP (Prajwal et al., 2020) synchronize synthetic audio with authentic footage, producing highly believable fake speeches that preserve lip movements as well as subtle facial expressions and gestures. Real-time manipulation tools like FACE2FACE (Thies et al., 2016) enable alteration of facial expressions during live video calls, allowing impersonation of individuals such as CEOs or officials for fraudulent purposes.

Compared to single-modal forgeries, multi-modal methods provide several advantages. They create the impression of corroboration across modalities, which raises perceived authenticity. They also better avoid detection since inconsistencies may be hidden in the less examined modality. Moreover, multi-modal content tends to attract greater social media engagement, enabling faster spread. Lastly, the coherent presentation across modalities reduces cognitive load for users, making it easier for false information to be accepted and shared.

### 3.2 Formulation of Real-versus-Fake Image Forgery Detection

The rapid growth of digital media and forgery techniques has made it harder to verify image content authenticity. With easy access to advanced editing tools and realistic AI-generated media, it is often difficult to tell real images from fake ones, especially in cases of disinformation, identity theft, or evidence manipulation. These risks highlight the urgent need for automated, robust detection systems that can keep up with evolving forgeries.

Real-versus-fake classification forms the foundation of conventional forgery detection, aiming to determine whether a given image is authentic or has been synthetically altered. This task is typically framed as a binary classification problem. Let the training dataset be defined as $\mathcal{D} = \{(\mathbf{x}_n, y_n)\}_{n=1}^{N}$, where $\mathbf{x}_n \in \mathcal{X}$ denotes the input sample (*e.g.,* image, video), $N$ is the number of training samples, and $y_n \in \{0, 1\}$ is the authenticity label, with 0 indicating real and 1 indicating fake content, we denote the sets of real and fake samples as $\mathcal{R}$ and $\mathcal{F}$, respectively. A model $f_\theta : \mathcal{X} \to \{0, 1\}$, parameterized by $\theta$, processes the input samples. The objective is to minimize the loss $\mathcal{L}$:

$$\theta^* = \arg\min_\theta \frac{1}{N} \sum_{n=1}^{N} \mathcal{L}\big(f_\theta(\mathbf{x}_n), \ y_n\big) \tag{1}$$

As discussed earlier, forged content was mainly created using traditional methods (*e.g.,* splicing, copy-move, *etc.*) or early AI-driven techniques (*e.g.,* VAEs and GANs). The rise of diffusion models, such as STABLE DIFFUSION (CompVis, 2023) and IMAGEN (Saharia et al., 2022), has greatly improved the realism and variety of synthetic images, making them a key source of modern forgeries. To cope with this evolving challenge of detecting highly realistic diffusion-generated content that exhibits minimal traditional artifacts, some works have developed detection methods that exploit the unique properties of the diffusion process. For example, DIRE (Wang et al., 2023) and SEDID (Ma et al., 2023a) utilize discrepancies between forward and reverse generation steps to reveal subtle artifacts, while LATENTTRACER (Wang et al., 2024) operates in the latent space to trace generation trajectories, enabling effective detection even when the specific generative model is partially known. These approaches can be generally formulated as $f_\theta = ||x_0 - \hat{x}_\theta(x_t, t)||$, with $t$ representing the timestep in the diffusion process and $\hat{x}_\theta$ denoting the denoising network. Although diffusion models currently dominate, generative techniques continue to evolve, and future forgeries may stem from more diverse or hybrid generators. This presents the additional challenge of developing detection methods that can generalize across unknown and emerging generative architectures. Consequently, some researchers have explored model-agnostic detection strategies. ESSP (Chen et al., 2024a) boosts generalization to unseen generators by exploiting camera-specific noise retained in "simple patches", formulated as $f_\theta = \theta(\mathrm{SRM}(x))$, where SRM (Steganalysis Rich Model) (Fridrich & Kodovsky, 2012) uses high-pass filters to extract high-frequency noise patterns, which generative models tend to overlook in favor of textured regions. Meanwhile, UNIFD (Ojha et al., 2023) introduces a training-free method that detects forgeries via nearest-neighbor search, expressed as $f_\theta = d(\theta(x), \mathcal{R}) - d(\theta(x), \mathcal{F})$, where $\theta(x)$ represents CLIP feature extraction and $d(\cdot, \cdot)$ measures distances to the reference sets $\mathcal{R}$ and $\mathcal{F}$, offering high adaptability without model-specific assumptions. Furthermore, several approaches leverage cross-modal cues, particularly the semantic alignment between textual descriptions and images. Such methods are especially effective for detecting content generated by text-to-image models or forged image–text pairs. For example, FATFORMER (Liu et al., 2024b) explicitly integrates text–image alignment and incorporates text-guided semantic features, which can be formulated as $f_\theta = \cos\big(f_\theta^I(x^I), \ f_\theta^T(x^T)\big)$, where $f_\theta^I(\cdot)$ and $f_\theta^T(\cdot)$ denote modality-specific encoders that project images and text into a shared embedding space. By exploiting cross-modal consistency, FATFORMER achieves improved detection performance and is well aligned with the growing trend of text-driven generation.

While the aforementioned detection methods have achieved considerable progress in terms of accuracy and robustness, they predominantly operate as black-box systems with limited explainability capabilities. Overall, these approaches span diverse methodologies and application domains. Some specialize in identifying diffusion-based forgeries, others aim for general-purpose AI image detection, while certain methods are tailored for specific scenarios such as facial forgeries or text-to-image content. These methods have demonstrated notable advances in performance and efficiency. However, most remain limited to binary classification, outputting only a `"real"` or `"fake"` label without providing insights into the reasoning behind their decisions.

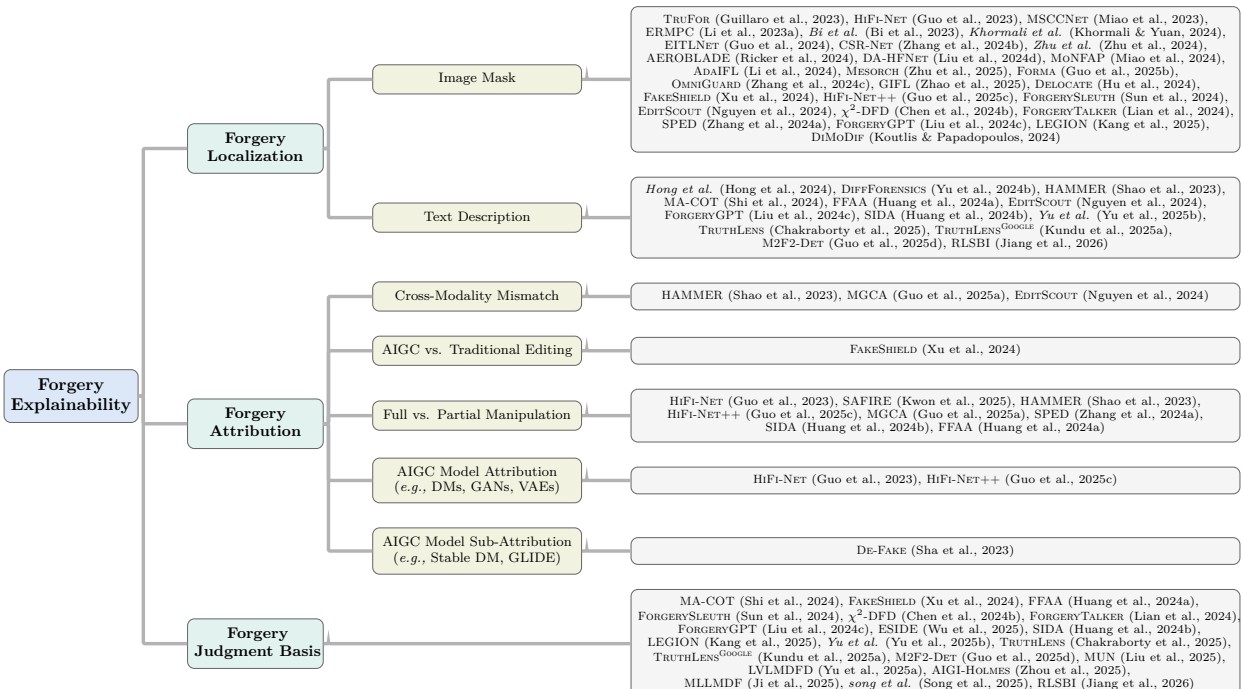

Figure 2: A hierarchical taxonomy of forgery explainability techniques. **Cross-Modality Mismatch:** Assessment of semantic consistency across different modalities (*e.g.,* image-text and image-audio alignment). **AIGC vs. Traditional Editing:** Attribution based on the source of manipulation (*e.g.,* AI-generated content versus traditional editing approaches such as Photoshop). **Full vs. Partial Manipulation:** Characterization of the extent of content manipulation (*e.g.,* fully manipulated images, partial modifications, face swaps, or object-level edits). **AIGC Model Attribution:** Attribution to broad categories of generative model families (*e.g.,* GANs, diffusion models, or VAE-based architectures). **AIGC Model Sub-Attribution:** Fine-grained attribution to specific generative models within a given category (*e.g.,* STABLE DIFFUSION (CompVis, 2023), GLIDE (Nichol et al., 2021), *etc.*).

This black-box nature limits their applicability in high-stakes domains such as forensic analysis, news verification, and content moderation, where explainability and traceability are crucial. To bridge this gap, this survey focus on the emerging field of explainable forgery detection, systematically reviewing its key explainable tasks. As illustrated in Figure 2, explainable detection typically comprises three core components:

- Forgery Localization: Identifying the specific manipulated regions within an image, either as pixel-level masks or textual descriptions.

- Forgery Attribution: Determining the type or source of manipulation, organized hierarchically from coarse to fine levels, ranging from broad categories such as cross-modality mismatch to specific model-level attribution.

- Forgery Judgment Basis: Explaining why an image is classified as fake by providing transparent and verifiable supporting evidence.

This taxonomy provides a structured foundation for advancing explainable forgery detection and facilitates the development of methods that go beyond binary decisions.

## 4 Explainable Forgery Detection

While concurrent work (Zou et al., 2025) briefly touches on explainability in AI-generated content detection, our work specifically focuses on the emerging paradigm of explainable forgery detection. We systematically

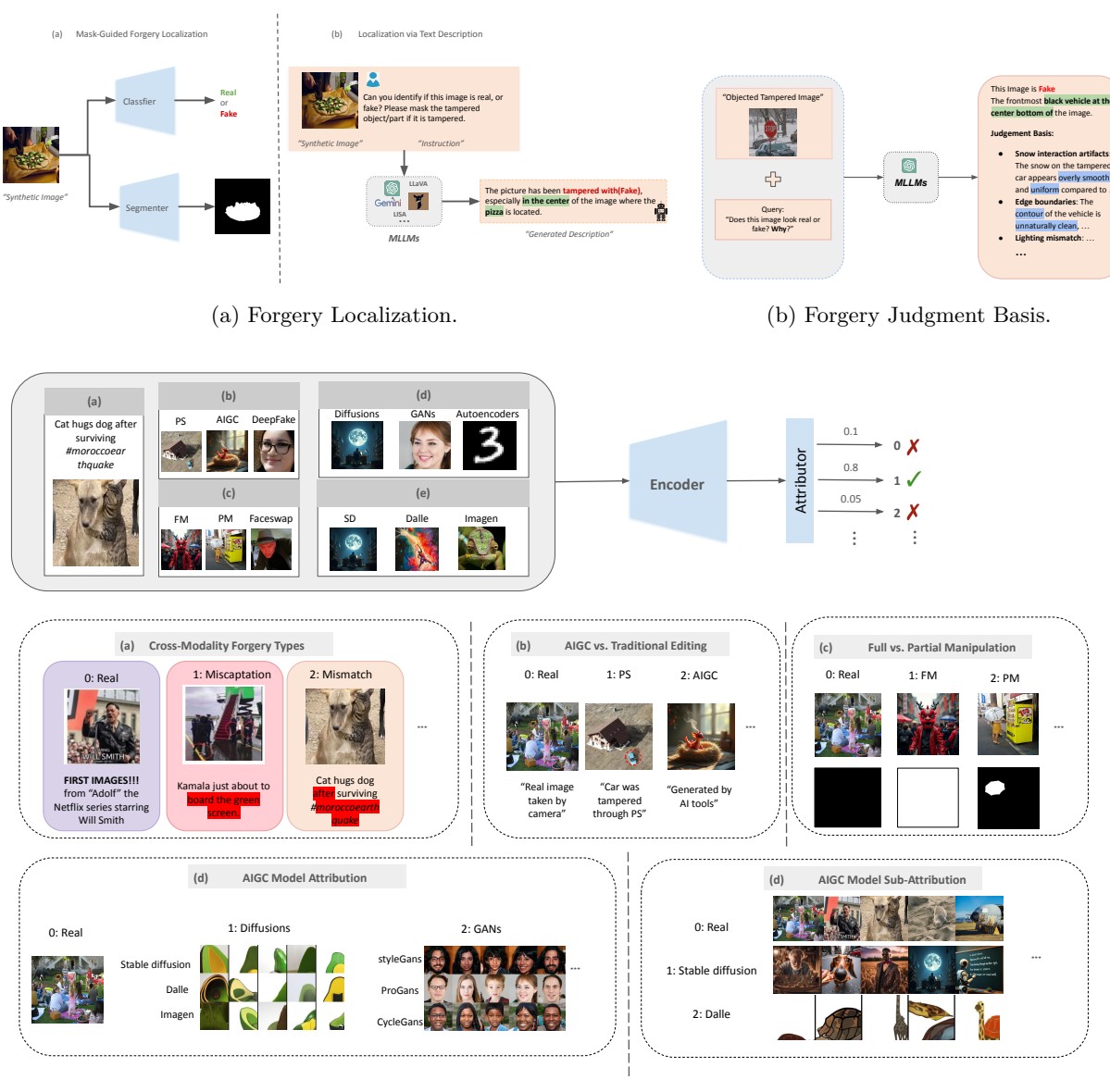

Figure 3: Illustration of three key explainability tasks in forgery detection: (a) Localization, (b) Judgment Basis, and (c) Attribution. (a) Shows mask-guided and text-based localization approaches for identifying tampered regions. (b) Demonstrates structured reasoning with specific evidence for detection decisions. (c) Presents hierarchical attribution from cross-modal mismatch to fine-grained model identification.

examine explainability requirements and establish a comprehensive taxonomy of explainable tasks. This section presents a structured framework that goes beyond binary classification to provide explainable analysis through three core dimensions: Forgery Localization (Section 4.1), which identifies manipulated regions within content; Forgery Attribution (Section 4.2), which determines the source and method of manipulation; and Forgery Judgment Basis (Section 4.3), which provides reasoning for detection decisions.

## 4.1   Forgery Localization

Forgery Localization (FL) aims to identify manipulated regions within an image by providing spatial localization, typically in the form of segmentation masks or natural language descriptions. Unlike traditional

binary classification methods that only output `"real"` or `"fake"`, FL addresses the crucial question of where the manipulation occurs. As illustrated in Figure 3a, FL generates image masks that highlight the tampered areas, while these masks demonstrate localization through textual descriptions specifying the affected regions and manipulation characteristics. This capability substantially enhances the explainability of detection results, fosters user trust and confidence, and provides valuable visual and semantic evidence for forensic analysis and content verification in high-stakes real-world scenarios.

Formally, given an input $X$ (either a single image $I \in \mathbb{R}^{H \times W \times C}$ or a video sequence), visual localization methods produce outputs such as:

- Pixel-wise probability map: $P \in [0,1]^{H \times W}$, where $P(i,j)$ indicates the probability that pixel $(i,j)$ has been manipulated.

- Instance masks: $M = \{M_1, \ldots, M_k\}$, where each $M_i \in \{0,1\}^{H \times W}$ corresponds to a binary mask of an individual tampered region, enabling fine-grained segmentation akin to instance segmentation techniques (*e.g.,* SAM (Kirillov et al., 2023b)).

- Bounding boxes: $B = \{B_1, \ldots, B_k\}$, where each $B_i = [x_i^{\text{center}}, y_i^{\text{center}}, w_i, h_i]$ represents a rectangular region around suspected manipulations. Here, $(x_i^{\text{center}}, y_i^{\text{center}})$ denotes the center coordinates and $w_i, h_i$ denote the width and height, respectively.

- Frame-level binary maps: $F \in \{0,1\}^{T \times D}$, where $T$ denotes the number of video frames and $D$ represents the number of modalities (*e.g.,* visual and audio). Each entry $F_{t,d} = 1$ indicates that frame $t$ contains manipulated content in modality $d \in \{v, a\}$, enabling coarse-grained temporal localization across different media streams. For example, $F_{1,v} = 1$ and $F_{1,a} = 0$ means that the 1st frame is visually manipulated but has authentic audio.

In addition, textual localization, particularly in multimodal settings, generates natural language descriptions $T$ that explicitly specify manipulated regions, *e.g.,* "The mouth region in the face is fake". We can thus define a localization model $\Lambda_\phi$ as a mapping $\Lambda_\phi : X \mapsto \mathcal{O}$, where the output space $\mathcal{O}$ includes one or more forms:

$$\mathcal{O} \subseteq \{P, M, B, F, T\} \tag{2}$$

Here, $T$ represents textual output describing manipulated areas within the input. Such flexible and multi-form outputs allow localization models to suit a wide range of forgery detection scenarios, enhancing transparency and explainability. Most forgery localization methods can be categorized into five main approaches:

- Pixel-level mask methods offer the most precise spatial localization, especially in boundary identification. These methods typically predict pixel-wise tampering probability maps by fusing image representations with tampering features. For instance, TRUFOR (Guillaro et al., 2023) and MUN (Liu et al., 2025) integrate noise pattern analysis to generate probabilistic masks, HIFI-NET (Guo et al., 2023) leverages multi-scale feature fusion to output binary masks, MSCCNET (Miao et al., 2023) captures frequency-domain traces through a multi-spectral class center module, and AEROBLADE (Ricker et al., 2024) achieves fine-grained localization via reconstruction error from an autoencoder.

- Instance-level localization methods further incorporate semantic understanding. *Khormali et al.* (Khormali & Yuan, 2024) model facial images as graphs and combine features extracted by a Vision Transformer with a graph convolutional network and transformer-based discriminator to produce a relevancy map, allowing not only spatial localization but also identification of the manipulated facial components.

- Bounding box-based methods provide coarse localization using rectangular regions. For example, HAMMER (Shao et al., 2023) fuses image features from a Vision Transformer and text features from BERT through cross-modal attention to predict bounding boxes that enclose manipulated regions, offering more intuitive visual feedback.

- Frame-level localization extends explainability along the temporal dimension. DIMODIF (Koutlis & Papadopoulos, 2024) extracts features from visual and audio speech recognition and employs a transformer

Table 2: Pixel-level F1 performance comparison of forgery localization methods on commonly used benchmark datasets. A dash (–) indicates that the result was not reported in the original paper. **Bold** is the best and underline is the second best.

| Method | Feature-Driven Detection | CASIAv1+ | Coverage | Columbia | NIST16 | IMD20 | DSO-1 | CocoGlide |
|---|---|---|---|---|---|---|---|---|
| TruFor ( Guillaro et al. (2023)) | Noiseprint, RGB | 0.822 | 0.735 | 0.914 | 0.470 | – | **0.973** | 0.720 |
| ERMPC ( Li et al. (2023a)) | Noiseprint, RGB | **0.876** | **0.944** | **0.968** | **0.895** | **0.856** | – | – |
| EITLNet ( Guo et al. (2024)) | Noiseprint, RGB | 0.530 | 0.448 | 0.881 | 0.308 | 0.532 | – | 0.410 |
| CSR-Net ( Zhang et al. (2024b)) | Rep. | 0.585 | 0.780 | – | 0.835 | – | – | – |
| Zhu *et al.* ( Zhu et al. (2024)) | Noiseprint, RGB | 0.621 | 0.812 | – | 0.868 | – | – | – |
| DA-HFNet ( Liu et al. (2024d)) | Noiseprint, RGB, Freq. | – | – | – | – | – | – | 0.585 |
| AdaIFL ( Li et al. (2024)) | Rep., RGB | 0.848 | 0.745 | – | 0.706 | – | – | – |
| ForgeryGPT ( Liu et al. (2024c)) | Rep. | 0.569 | – | 0.773 | 0.549 | 0.530 | 0.506 | – |
| Forma ( Guo et al. (2025b)) | Freq. | 0.729 | 0.587 | 0.949 | 0.454 | – | – | 0.453 |
| GIFL ( Zhao et al. (2025)) | Rep. | 0.783 | 0.565 | – | – | – | – | 0.571 |
| FakeShield ( Xu et al. (2024)) | Rep. | 0.600 | – | 0.750 | 0.370 | 0.570 | 0.520 | – |
| ForgerySleuth ( Sun et al. (2024)) | Rep. | 0.870 | 0.792 | 0.931 | 0.610 | – | – | **0.751** |
| EditScout ( Nguyen et al. (2024)) | Rep. | – | – | – | – | – | – | 0.457 |

encoder to process cross-modal discrepancies. It combines classification and regression heads to identify manipulated segments in the video timeline.

- Text-based explanation methods generate natural language descriptions through fine-tuned large language models. Methods such as ForgerySleuth (Sun et al., 2024), FakeShield (Xu et al., 2024), ForgeryGPT (Liu et al., 2024c), and RLSBI (Jiang et al., 2026) adopt various strategies, including integrating the LLaVA (Liu et al., 2023) framework to generate text that explains the manipulation's location, type, and rationale, or using LoRA-tuned language models to generate analytical descriptions which are then linked to pixel-level masks via the SAM (Kirillov et al., 2023a) model, providing textual support for visual explanations.

These five categories reflect the current technical landscape of explainable forgery localization. Through diverse feature extraction and fusion strategies, they collectively enable multi-level explainable outputs ranging from fine-grained spatial localization to high-level semantic localization, supporting comprehensive understanding in forgery detection.

**Comparative Analysis of Localization Methods.** Table 2 reports pixel-level F1 scores of representative localization methods across seven benchmark datasets. Three key patterns emerge. First, dual-stream methods that fuse noise-based and RGB features achieve consistently strong performance on traditional manipulation datasets. ERMPC Li et al. (2023a) obtains the highest F1 on five benchmarks (CASIAv1+ Dong et al. (2013a): 0.876, Coverage Wen et al. (2016): 0.944, Columbia Ng et al. (2009): 0.968, NIST16: 0.895, IMD20 Novozamsky et al. (2020): 0.856), demonstrating the effectiveness of combining edge-aware message passing with noise residual extraction. TruFor achieves the best score on DSO-1 De Carvalho et al. (2013) (0.973) but drops notably on NIST16 Guan et al. (2019) (0.470), whose small forged regions and atypical scene content pose particular challenges for noise-driven approaches. Second, performance varies substantially across datasets, reflecting inherent difficulty differences. Columbia and CASIAv1+ Ng et al. (2009); Dong et al. (2013a), containing conventional splicing with clear boundary artifacts, yield relatively high scores across methods. In contrast, NIST16 (small forged regions) and Coverage (highly similar source and target regions) remain more challenging. On the diffusion-generated CocoGlide Guillaro et al. (2023) dataset, ForgerySleuth (0.751) and TruFor (0.720) outperform other methods by a large margin, while most approaches fall below 0.60, indicating that detecting AI-generated forgeries remains a significant challenge. Third, multi-modal methods incorporating large language models show a trade-off between semantic explainability and spatial precision. ForgerySleuth achieves competitive localization (*e.g.*, 0.870 on CASIAv1+ Dong et al. (2013a), 0.931 on Columbia Ng et al. (2009), 0.751 on CocoGlide Guillaro et al. (2023)), whereas FakeShield Xu et al. (2024) (0.370 on NIST16) and ForgeryGPT Liu et al. (2024c) (0.549 on NIST16 Guan et al. (2019)) exhibit lower precision, suggesting that sharing model capacity between mask prediction and text generation can reduce localization accuracy. We note that direct comparison should be interpreted with caution, as training data, augmentation, and threshold settings vary across studies. Methods such as HiFi-Net Guo et al. (2023), HiFi-Net++ Guo et al. (2025c), MSCCNet Miao et al. (2023), and MoNFAP Miao et al. (2024), which were evaluated under different splits or specialized condi-

tions (*e.g.*, JPEG-compressed inputs), are not included in Table 2. The lack of a unified protocol remains a barrier to fair comparison, as discussed in Section 7.

**Cross-Category Analysis of FL Approaches.** The remaining four FL paradigms (instance-level, bounding box, frame-level, and text-based) lack common benchmarks for quantitative comparison. Instance-level and bounding box methods have only one or two representatives each. Frame-level methods (*e.g.*, DiMoDif) use temporal metrics (AP@$p$, AR@$n$) incompatible with spatial IoU/F1. Text-based methods (*e.g.*, FORGERYSLEUTH Sun et al. (2024), FAKESHIELD Xu et al. (2024), FORGERYGPT Liu et al. (2024c), LE-GION Kang et al. (2025)) rely on NLP metrics (BLEU, ROUGE-L, CIDEr) that measure linguistic quality rather than spatial precision. This evaluation heterogeneity highlights the absence of cross-paradigm benchmarks for explainable localization, a gap we address in Section 7.

## 4.2 Forgery Attribution

While forgery localization focuses on *where* manipulations occur within an image, Forgery Attribution (FA) addresses a complementary question: *how* the manipulation was performed. Rather than identifying spatial regions of tampering, FA aims to explain the semantic nature and technical origin of the manipulation. To support this goal, we categorize attribution into five progressively fine-grained levels, each capturing a specific perspective on the manipulation process, as shown in Figure 3c. At the coarsest level, Cross-Modality Mismatch assesses whether information conveyed across different modalities, such as image-text or image-audio pairs, is semantically consistent. This capability is particularly important in multimodal disinformation scenarios, where content coherence is intentionally disrupted. The second level, AIGC *vs.* Traditional Editing, distinguishes between manipulations generated by AI models (such as diffusion models or GANs) and those produced using conventional editing tools (such as splicing or inpainting via Photoshop). This classification provides insight into the generation pipeline and the associated risk profile. The third level, Full vs. Partial Manipulation, characterizes the manipulation scope by determining whether the entire image is synthetic or only specific regions have been modified, such as in localized face swaps or object removal. This distinction helps assess the semantic severity and visual impact of the forgery. Moving further, AIGC Model Attribution identifies the category of generative model responsible for the manipulation, for example GANs, VAEs, or diffusion models. This level supports model family traceability and enables the detection of model-specific artifacts. Finally, AIGC Model Sub-Attribution further refines the analysis by pinpointing the exact generative model instance used, such as Stable Diffusion or GLIDE. This fine-grained attribution is valuable for forensic accountability, provenance tracking, and validation of anti-forgery measures such as watermarks.

Building on this intuition, forgery attribution can be formally modeled as a multi-class classification problem that goes beyond binary judgments. Specifically, we define a set of content categories $\mathcal{C} = \{c_0, c_1, \ldots, c_n\}$, where $c_0$ represents `real` content, and $\{c_1, \ldots, c_n\}$ correspond to different manipulation types, including synthetic generations and manual tampering. A model $f_\psi : \mathcal{X} \to \mathcal{C}$ is trained to predict the manipulation category of each input sample:

$$\psi^* = \arg\min_\psi \frac{1}{N} \sum_{n=1}^{N} \mathcal{L}_{\text{ce}}\big(f_\psi(x_n),\ c_n\big) \tag{3}$$

where $\mathcal{L}_{\text{ce}}$ denotes the standard cross-entropy loss. This formulation provides a unified way to accommodate coarse and fine-grained attribution schemes under a common framework.

Based on the above multi-class formulation, existing methods define different label spaces $\mathcal{C}$ to model various aspects of manipulation, thereby enabling more structured and explainable forgery attribution. For cross-modality mismatch, MGCA (Guo et al., 2025a) extracts semantic elements from image-text pairs, such as entities and temporal cues, and aligns them to detect inconsistencies. HAMMER (Shao et al., 2023) constructs real and tampered multimodal pairs and applies contrastive learning to capture semantic shifts induced by manipulation. These models allow outputs that indicate not only whether manipulation occurred, but also where semantic conflicts arise across modalities. To distinguish AIGC from traditional editing, FAKESHIELD (Xu et al., 2024) treats manipulation sources such as AIGC, DeepFake, and Photoshop as classification targets. It generates manipulation-type prompts using a domain tag predictor and feeds them into a multimodal model to assist in identifying the editing mechanism. When modeling full

versus partial manipulation, HIFI-NET++ (Guo et al., 2025c) and HAMMER (Shao et al., 2023) first assess image authenticity, then localize tampered regions to estimate manipulation extent. SAFIRE (Kwon et al., 2025) uses the Segment Anything Model to propose candidate regions and assesses source consistency, while MGCA (Guo et al., 2025a) segments manipulated areas using class-guided features. SIDA (Huang et al., 2024b) and FFAA (Huang et al., 2024a) generate both manipulation masks and descriptive text to support spatial attribution, and SPED (Zhang et al., 2024a) detects frequent forgery artifacts to assist in partial manipulation recognition. To identify the type of generative model involved, HIFI-NET (Guo et al., 2023) and HIFI-NET++ (Guo et al., 2025c) classify images based on features left by different model categories such as GANs or diffusion models. HIFI-NET++ (Guo et al., 2025c) introduces a hierarchical structure to first detect AIGC-generated content and then determine its model family, enabling more accurate and explainable predictions. For finer-grained model sub-attribution, DE-FAKE (Sha et al., 2023) maps fake images to specific generators like Stable Diffusion or GLIDE (Nichol et al., 2021). It designs both image-only and image-plus-prompt classifiers to capture subtle model-specific signals and improve instance-level discrimination. By treating manipulation types, regions, and source models as structured outputs, forgery attribution enhances the link between model representations and forgery semantics. This supports improved explainability through attention maps, concept-level reasoning, and traceable model behavior.

### 4.3 Forgery Judgment Basis

Unlike Forgery Localization (FL), which focuses on identifying *where* manipulations occur, and Forgery Attribution (FA), which explains *how* and *by what means* forgeries are created, Forgery Judgment Basis (FJB) addresses the fundamental question of *why* a detection model determines an image to be fake. In other words, it focuses on what visual evidence the model relies on to support its decision. FJB requires detection systems to output not only classification results but also explicit, human-explainable reasoning based on observable artifacts in the image. As illustrated in Figure 3b, when presented with a suspected tampered image, an FJB-enabled system provides structured justifications across multiple semantic dimensions. The system identifies the manipulated object or region, such as "the frontmost black vehicle at the bottom center of the image", and presents detailed reasoning elements such as snow interaction inconsistencies, where "the snow on the tampered car appears overly smooth and uniform compared to the surrounding terrain", unnatural edge boundaries, where "the contour of the vehicle is excessively clean and lacks natural blending", and scene-level lighting mismatches. These explanations should be both readable and verifiable, helping users understand which regions the model distrusts and what evidence leads to that assessment. The motivation for FJB lies in a key limitation of current detection systems. Although many models can achieve high classification accuracy, they typically do not reveal the internal reasoning behind their predictions. This lack of transparency becomes a significant barrier in high-stakes applications such as forensic analysis, regulatory inspection, and journalistic verification, where explanations must be evidence-based and traceable. By linking detection results to specific, explainable image cues, FJB improves the transparency, debuggability, and accountability of image forgery detection models.

To formalize this process, we define FJB as the task of extracting explainable reasoning factors that substantiate a model's authenticity judgment. These factors may include texture inconsistencies, edge artifacts, lighting or shading anomalies, semantic contradictions, or temporal mismatches across modalities (*e.g.*, audio-visual misalignment). Given an input sample $X$ and its classification result $f(X; \theta)$, the explanation module $g$ aims to generate a set of explicit reasoning elements that support the decision:

$$g(f(X; \theta), X; \omega) = R \tag{4}$$

where $R = \{r_1, r_2, \ldots, r_k\}$ represents the set of judgment elements supporting the decision, and $\omega$ denotes the parameters of the explanation module. In practice, this task is often approached through fine-tuning large-scale models to produce textual justifications or by training on datasets with annotated rationale. Outputs may include attention maps, saliency-based attributions, or natural language explanations, all of which provide insight into the model's internal decision-making process.

Building on the formulation above, existing FJB methods typically generate natural language explanations to reveal *why* a model considers an image to be manipulated. One group of approaches employs prompt-based strategies to explicitly guide LLMs toward structured justifications. For example, FORGERYSLEUTH

(Sun et al., 2024) decomposes the reasoning process into sub-tasks such as identifying manipulated objects, explaining specific evidence, and articulating conclusions, using separate prompts to generate answers for each stage. FORGERYGPT (Liu et al., 2024c) adopts a unified question-answering format to elicit complete textual rationales in a single step. Other methods enhance controllability by conditioning the generation on auxiliary semantic cues. FAKESHIELD (Xu et al., 2024) first predicts the manipulation type (*e.g.,* AIGC or Photoshop), then uses the predicted label as a domain-specific prompt to generate type-aware explanations. $\chi^2$-DFD (Chen et al., 2024b) performs statistical comparisons between real and fake images and converts the results into textual templates for explanation. Some models further align textual output with internal model representations. *Yu et al.* (Yu et al., 2025b) extract manipulation-aware visual features and feed them into the language model as conditioning inputs, improving alignment between explanation content and model attention. FORGERYTALKER (Lian et al., 2024) detects manipulated regions and encodes them as region-specific keywords, which are used to prompt a language model to generate localized textual explanations for why those regions appear fake. SIDA (Huang et al., 2024b) and FFAA (Huang et al., 2024a) incorporate dual learning mechanisms to enforce consistency between generated explanations and model-predicted manipulation masks, improving spatial grounding of the output. Similarly, LEGION (Kang et al., 2025) and M2F2-DET (Guo et al., 2025d) integrate explanation generation into multimodal understanding modules, jointly predicting salient regions and corresponding textual evidence. Building on these multimodal paradigms, recent methods leverage large-scale vision-language models: LVLMDFD (Yu et al., 2025a) fine-tunes MLLMs with forgery-specific prompts, AIGI-HOLMES (Zhou et al., 2025) employs preference optimization for human-aligned explanations, MLLMDF (Ji et al., 2025) adopts training-free prompting strategies, *Song et al.* (Song et al., 2025) design multi-paradigm reasoning frameworks, and RLSBI (Jiang et al., 2026) applies specialized prompt engineering for deepfake scenarios.

Other approaches adopt a more structured strategy by constructing explanation templates or tag-based descriptions. MA-COT (Shi et al., 2024) performs multi-attribute analysis across various forgery cues (*e.g.*, lighting, texture, boundary), then aggregates the results and reformulates them into coherent textual rationales. ESIDE (Wu et al., 2025) and TRUTHLENS (Chakraborty et al., 2025) identify salient visual regions or concept-level forgery artifacts and express them as concise textual descriptions using predefined patterns. Although these methods vary in design, they share a common objective which is to generate human-readable and verifiable explanations that connect model predictions with explainable reasoning.

## 5 Forgery Detection Strategies

While Section 4.3 establishes *what* explainable forgery detection aims to achieve through FL, FA, and FJB, this section focuses on the feature-driven strategies that support these objectives. Rather than treating feature extraction as an isolated design choice, we examine it from an explainability perspective by asking what kinds of explainable outputs each feature paradigm can provide and what limitations it entails. As illustrated in Figure 4, existing methods mainly rely on four types of features: RGB, frequency, noiseprint, and representation. These paradigms span from low-level pixel statistics to high-level semantic abstractions, and each offers different strengths for different explainability tasks. Single-modal methods typically exploit low-level forensic cues, such as edge inconsistencies or spectral anomalies, within a single modality. By contrast, multi-modal methods focus on semantic consistency across modalities, such as image–text or audio–visual pairs, and therefore rely primarily on representation features to capture cross-modal discrepancies. This is largely because multi-modal forgeries often manifest as semantic inconsistencies rather than obvious pixel-level artifacts, making a shared embedding space more suitable for comparison and reasoning. Section 5.5 builds on these observations by systematically mapping each feature type to the explainability tasks it supports, based on the empirical evidence summarized in Table 3. The table summarizes representative methods from both paradigms in terms of targeted manipulation types, architectural design, feature-driven strategy, explainability output, and code availability.

### 5.1 RGB Features

RGB features serve as the most direct and widely used input modality in visual forgery detection. They retain complete pixel-level spatial and color information of the image without requiring additional transformations

Table 3: Overview of architecture and feature strategies for forgery detection methods.

| Method | Target Synthetic Images† | Strategies‡ | Output§ | Backbones Vision | Non-Vision | Multi-Modality | Feature-Driven Detection | Real vs. Fake | Exp.¶ | Code Availability |
|---|---|---|---|---|---|---|---|---|---|---|
| **Single-Modality** | | | | | | | | | | |
| **Image** | | | | | | | | | | |
| TruFor (Guillaro et al., 2023) | T: SP, CM | SSL, SL | M | DnCNN, SegFormer | - | - | Noiseprint, RGB | ✓ | FL | ✓ |
| HiFi-Net (Guo et al., 2023) | T: SP G: DMs | SL | C, M | ResNet50 | - | - | RGB, Freq. | ✓ | FL, FA | ✓ |
| MSCCNet (Miao et al., 2023) | G: GANs, DMs | SL | C, M | ResNet50 | - | - | Freq. | ✓ | FL | ✓ |
| ERMPC (Li et al., 2023a) | T: SP, CM, RM | SL | M | ResNet50 | - | - | Noiseprint, RGB | ✗ | FL | ✗ |
| Bi et al., (Bi et al., 2023) | T: SP, CM, RM | SSL | M | CNN-based | - | - | Noiseprint | ✗ | FL | ✗ |
| Khormali et al., (Khormali & Yuan, 2024) | T: Low Resolution, Color Mismatch, Inaccurate Fake Masks G: DeepFakes, Face2Face, FaceSwap, NeuralTextures | SSL | C, M | ViT, Graph CNN | - | - | Rep. | ✓ | FL | ✗ |
| EITLNet (Guo et al., 2024) | T: SP, CM, RM | SL | M | SegFormer | - | - | Noiseprint, RGB | ✗ | FL | ✓ |
| CSR-Net (Zhang et al., 2024b) | T: SP, CM, RM | SL | M | ResNet50 | - | - | Rep. | ✗ | FL | ✓ |
| Zhu et al., (Zhu et al., 2024) | T: SP, CM, RM | SL | C, M | ResNet50 ConvGeM | - | - | Noiseprint, RGB | ✓ | FL | ✗ |
| AEROBLADE (Ricker et al., 2024) | G: LDMs | TF | C | - | - | - | Rep. | ✓ | FL | ✓ |
| DA-HFNet (Liu et al., 2024d) | G: Img/Text-Guided GANs, DMs | SL | C, M | HRNet | - | - | Noiseprint, RGB, Freq. | ✓ | FL | ✗ |
| Hong et al., (Hong et al., 2024) | T: Low Resolution, Color Mismatch, Inaccurate Fake Masks G: DeepFakes, Face2Face, FaceSwap, NeuralTextures | SL | T | ResNet34, ResNet50, SwinT | - | - | Rep. | ✓ | FL | ✗ |
| DiffForensics (Yu et al., 2024b) | T: SP, CM, RM G: AI-Editing | SSL, SL | T | SegFormer, DDPM | - | - | Rep., Freq. | ✓ | FL | ✗ |
| MoNFAP (Miao et al., 2024) | G: GANs, FaceSwap, FsGAN, DEEPFaceLab | SL | M | ConvNeXtV2-atto+ ResNet50 | - | - | RGB, Freq. | ✓ | FL | ✓ |
| AdaIFL (Li et al., 2024) | T: SP, CM, RM | SL | M | ViT | - | - | Rep., RGB | ✗ | FL | ✓ |
| SAFIRE (Kwon et al., 2025) | T: CM, SP G: DMs, SAM | SL | C, T | SAM | - | - | Rep. | ✗ | FA | ✓ |
| Mesorch (Zhu et al., 2025) | T: SP, CM, RM | SL | M | {ResNet50, ConvNeXT-Tiny} + {MAE, PvT0-B3, SegFormer-B3, SwinT-Base} | - | - | Rep., RGB, Freq. | ✗ | FL | ✓ |
| Forma (Guo et al., 2025b) | T: SP, CM, RM | SL | M | VMamba-tiny | - | - | Freq. | ✗ | FL | ✓ |
| OmniGuard (Zhang et al., 2024c) | G: VAEs, SDs | SL | M | VAE, ViT-B | - | - | Rep. | ✗ | FL | ✗ |
| GIFL (Zhao et al., 2025) | T: SP, CM, RM G: Deepfillv2, CTSDG, CR-Fill, LaMa, LDM, SSDE, DDNM, Repaint | SL | M | ViT-L/14 | - | - | Rep. | ✗ | FL | ✓ |
| MUN (Liu et al., 2025) | T: SP, CM, RM G: DMs, GANs, AEs,Deepfakes | SL | C, M | ConvNeXt V2 | - | - | Noiseprint, RGB | ✓ | FL | ✗ |
| **Video** | | | | | | | | | | |
| DeLocate (Hu et al., 2024) | T: Low Resolution, Color Mismatch, Inaccurate Fake Mask G: AEs, GANs, DEEPFakes | SL | M | ViT, ResNet-18 | - | - | Rep. | ✓ | FL | ✗ |
| **Multi-Modality** | | | | | | | | | | |
| **Image-Text** | | | | | | | | | | |
| De-Fake (Sha et al., 2023) | G: SDs | SL | C | ViT-B/32 | Transformer | MLP | Rep. | ✓ | FA | ✓ |
| HAMMER (Shao et al., 2023) | T: Misinformation | SL | C, T | ViT-B/16 | BERT-base | Attention + MLP | Rep. | ✓ | FL, FA | ✓ |
| MA-COT (Shi et al., 2024) | G: Face2Face, FaceSwap, NeuralTextures DeepFakes, GANs, DMs | TF | T | - | - | GPT-4v, Gemini | - | ✓ | FL, FJB, FA | ✓ |
| FakeShield (Xu et al., 2024) | T: PS G: DMs, Deepfakes | SL | C, M, T | ViT-L/14, SAM | - | LLaVA | Rep. | ✓ | FL, FJB, FA | ✓ |
| HiFi-Net++ (Guo et al., 2025c) | G: DMs, GANs, Faceshifter | SL | C, M | FPN, ResNet50 | Transformer | - | Rep., RGB, Freq. | ✓ | FL, FA | ✓ |
| FFAA (Huang et al., 2024a) | G: DeepFakes, Face2Face, FaceSwap, GANs, DMs | SL | T | ViT-L/14 | T5-Encoder | LLaVA-v1.6-7B | Rep. | ✓ | FL, FJB, FA | ✓ |
| ForgerySleuth (Sun et al., 2024) | T: SP, CM, RM, PS | SL | M, T | ViT-H | - | LLaVA-v1-7B | Rep. | ✓ | FL, FJB | ✓ |
| EditScout (Nguyen et al., 2024) | G: DMs | SL | M, T | SAM | - | LLaVA | Rep. | ✗ | FL | ✗ |
| χ²-DFD (Chen et al., 2024b) | G: DeepFakes, Face2Face, FaceSwap, NeuralTextures | SL | C, M, T | - | GPT-4o | LLaVA-7B | Rep. | ✓ | FL, FJB, FA | ✗ |
| MGCA (Guo et al., 2025a) | T: Misinformation | SL | C | ViT-B/16 | Vicuna, BERT | LLaVA-1.5 | Rep. | ✓ | FA | ✓ |
| ForgeryTalker (Lian et al., 2024) | G: GANs, DMs-Inpainting | SL | M, T | ViT, CoordConv Block with MHA | FlanT5/Vicuna | Q-Former | Rep. | ✗ | FL, FJB | ✗ |
| SPED (Zhang et al., 2024a) | T: Warped Photo Attacks, Cut Photo Attacks, Replay/Video Attacks | TF | C, M | Unspecified* | Unspecified* | - | - | ✓ | FL | ✗ |
| ForgeryGPT (Liu et al., 2024c) | T: SP, CM, RM G: Repaint-Inpaint(SAM, DMs) | SL | C, M, T | ViT-B/16 | Vicuna-7B | - | Rep. | ✓ | FL, FJB, FA | ✗ |
| ESIDE (Wu et al., 2025) | G: DMs | SL | C, T | DDIM, ViT-L/14 | GPT-4o | - | Rep., Freq. | ✓ | FL, FJB | ✓ |
| SIDA (Huang et al., 2024b) | G: DMs | SL | C, M, T | - | - | Lang-SAM, LISA | Rep. | ✓ | FL, FJB, FA | ✓ |
| LEGION (Kang et al., 2025) | G: DMs | SL | C, M, T | ViT-H/14 , SAM | - | GLaMM | Rep. | ✓ | FL, FJB | ✓ |
| Yu et al., (Yu et al., 2025b) | G: DeepFakes, Face2Face, FaceSwap, NeuralTextures | SL | C, T | ImageBind-Huge | ImageBind-Huge | Vicuna-7B | Rep. | ✓ | FL, FJB | ✗ |
| TruthLens (Chakraborty et al., 2025) | G: GANs, DMs | TF | T | - | - | LLaVA, BLIP-2, GPT-4 | - | ✓ | FL, FJB, FA | ✗ |
| TruthLens^Google (Kundu et al., 2025a) | G: DeepFakes, Face2Face, FaceSwap, NeuralTextures, DMs | SL | T | SigLIP-So400m/14, DINOv2 | Gemma2-3B | PaliGemma2 | Rep. | ✓ | FL, FJB, FA | ✗ |
| M2F2-Det (Guo et al., 2025d) | G: DeepFakes, Face2Face, FaceSwap, NeuralTextures | SL | C, T | ViT-L/14 , EfficientNet-B4 | Transformer | Vicuna-7B | Rep. | ✓ | FL, FJB | ✓ |
| LVLMDFD (Yu et al., 2025a) | G: DeepFakes | SL | C, T | - | - | LLaVA, MiniGPT-4, BLIP-2 | Rep. | ✓ | FJB | ✓ |
| AIGI-Holmes (Zhou et al., 2025) | G: DMs, GANs, DeepFakes | SL | C, T | ViT-L/14 | - | LLaVA | Rep. | ✓ | FJB | ✓ |
| MLLMDF (Ji et al., 2025) | G: DMs, GANs, AEs | TF | C, T | - | - | GPT-4o, LLaVA | - | ✓ | JB | ✓ |
| Song et al (Song et al., 2025) | G: DMs, GANs, VAEs | SL | C, T | - | - | GPT-4o, GPT-4o-mini, Llama-3.2, LLaVA | Rep. | ✓ | JB | ✓ |
| RLSBI (Jiang et al., 2026) | G: DeepFakes | SL | C, T | - | - | LLaVA-1.5 | Rep. | ✓ | FL, FJB | ✓ |
| **Image-Audio** | | | | | | | | | | |
| DiMoDif (Koutlis & Papadopoulos, 2024) | G: GANs, Faceswap, TTS, Wav2Lip, ChatGPT | SL | M | VSR | ASR | Transformer | Rep. | ✓ | FL | ✓ |

† **T:** Traditional editing methods, *e.g.*: Splicing(SP), Copy and Move(CM), Removal(RM); **G:** GAN- and Diffusion-based synthetic images.
‡: SL = Supervised Learning, SSL = Self-Supervised Learning, TF = Training Free.
§: M = Mask, C = Class, T = Text.
¶: RF = Real vs Fake, FL = Localization, FJB = Forgery Judgement Basis, FA = Attribution.
*: Backbone mentioned but not disclosed.
• References of mentioned synthetic-image generators in Table 3: DeepFakes (dee, 2019), Face2Face (Thies et al., 2016), FaceSwap (Kowalski, 2016), NeuralTextures (Thies et al., 2019), Stable Diffusion / LDMs (CompVis, 2023; Razzhigaev et al., 2023; Ramesh et al., 2022; Schuhmann et al., 2022), SAM (Kirillov et al., 2023a), FsGAN (Nirkin et al., 2022), DeepFaceLab (Perov et al., 2020), Deepfillv2 (Yu et al., 2019), CTSDG (Guo et al., 2021), CR-Fill (Zeng et al., 2021), LaMa (Suvorov et al., 2022), LDM (Rombach et al., 2022b), SSDE (Song et al., 2020b), DDNM (Wang et al., 2022b), Repaint (Lugmayr et al., 2022).
• References of mentioned backbones in Table 3: *Vision Backbones:* DnCNN (Zhang et al., 2017), SegFormer (Xie et al., 2021), ResNet50 / ResNet34 (He et al., 2016), CNN-based (Bi et al., 2023), ViT variants (B/16, B/32, L/14, H/14) (Dosovitskiy et al., 2020), GraphCNN (Kipf & Welling, 2016), SwinT / SwinT-Base (Liu et al., 2021), DDPM (Ho et al., 2020a), DDIM (Song et al., 2020a), ConvNeXtV2 (Woo et al., 2023), ConvGeM (Dong et al., 2022), HRNet (Sun et al., 2019), ConvNeXT-Tiny (Ma et al., 2023b), MAE (He et al., 2022), PvT0-B3 (Wang et al., 2021), VMamba-tiny (Liu et al., 2024e), FPN (Lin et al., 2017), EfficientNet-B4 (Tan & Le, 2019), ImageBind-Huge (Su et al., 2023), SAM (Kirillov et al., 2023a), CoordConv / MHA block (Liu et al., 2018), VSR (Ma et al., 2022). *Non-Vision Backbones:* Transformer (Vaswani et al., 2017b), BERT-base (Devlin et al., 2019), T5-Encoder (Raffel et al., 2020), GPT-4o (Achiam et al., 2023), ImageBind-Huge (Su et al., 2023), FlanT5 / Vicuna (InstructBLIP) (Dai et al., 2023), Gemma2-3B (Team et al., 2024), ASR (Ma et al., 2022) *Multi-Modality Modules:* GPT-4 / GPT-4v / GPT-4o (Achiam et al., 2023), Gemini (Team et al., 2023), LLaVA (Liu et al., 2023), LLaVA-v1.6 (Liu et al., 2024a), BLIP-2 (Li et al., 2023b), Q-Former (Dai et al., 2023), Vicuna-7B (Zheng et al., 2023), PaliGemma2 (Steiner et al., 2024), Lang-SAM (Team, 2024), LISA (Lai et al., 2024), GLaMM (Rasheed et al., 2024), SigLIP-So400m/14 (Alabdulmohsin et al., 2023), DINOv2 (Oquab et al., 2023), LLAMA-3.2 (Grattafiori et al., 2024).

or preprocessing. As illustrated in Figure 4, RGB features represent one of the primary shallow-level cues in feature-driven detection. They are extracted directly from raw image channels and preserve essential spatial and color structures for low-level visual analysis, including boundary inconsistencies, occlusion mismatches, and unnatural blending. Local visual anomalies introduced through region replacement or content splicing can often be identified using shallow convolutional networks applied directly to RGB inputs. The main advantage of RGB-based approaches lies in their computational efficiency and strong visual explainability, since detected anomalies can be directly observed and verified within the original image. However, when manipulation signals are subtle, such as compression artifacts or fine-grained noise inconsistencies, RGB

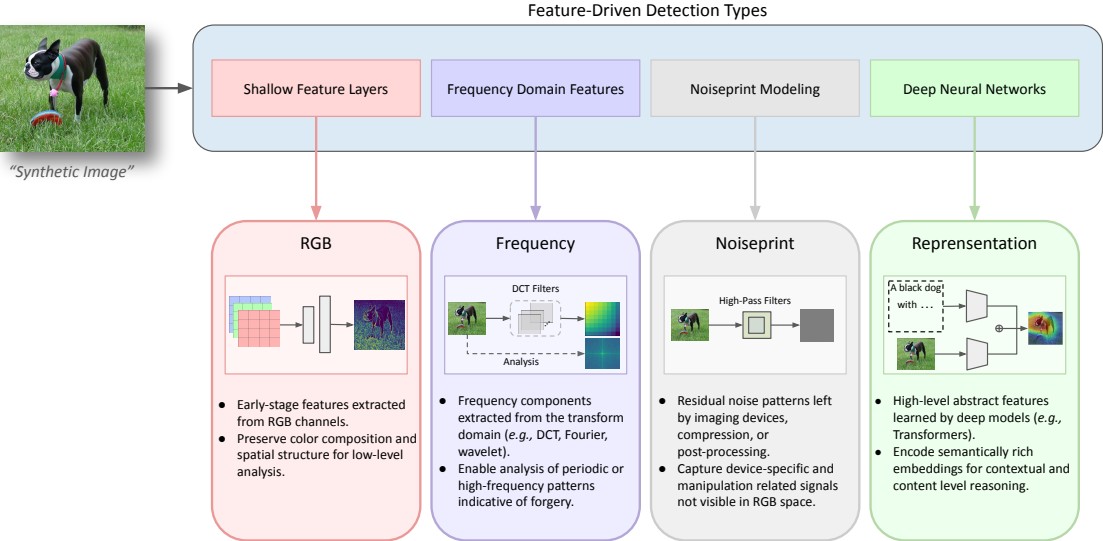

Figure 4: Feature-Driven types in forgery detection. Four primary feature extraction approaches: RGB Features preserve spatial and color information for boundary analysis, **Frequency Features** capture spectral anomalies through transform domain analysis (DCT, Fourier, wavelet), **Noiseprint Features** extract residual patterns via high-pass filtering to reveal device-specific traces, and Representation Features employ deep networks to learn semantic abstractions for multimodal reasoning.

features may prove insufficient because raw pixel intensities lack the ability to capture higher-order statistical variations or frequency-domain irregularities.

As summarized in Table 3, RGB-driven methods have adopted diverse architectural designs and enhancement strategies in unimodal detection settings. MONFAP (Miao et al., 2024) employs a hybrid backbone combining CONVNEXTV2-ATTO+ (Woo et al., 2023) and RESNET50 (He et al., 2016), integrating a noise-aware module to enhance multi-scale RGB features that highlight forgery-sensitive regions. DA-HFNET (Liu et al., 2024d) builds on HRNET (Sun et al., 2019) and incorporates channel and spatial attention mechanisms to emphasize discriminative RGB cues, particularly useful for detecting fine-grained local inconsistencies while preserving high-resolution spatial information. HIFI-NET (Guo et al., 2023) is based on a multi-branch RESNET50 (He et al., 2016) architecture, jointly processing RGB and frequency-domain features through hierarchical fusion, thus capturing both fine-scale manipulation traces and global consistency patterns. Some models treat RGB as an auxiliary modality. For example, ADAIFL (Li et al., 2024) leverages a ViT-based architecture that fuses RGB with high-level representation features to improve generalization across manipulation types. MESORCH (Zhu et al., 2025) combines ResNet50 with multiple Transformer variants (e.g., MAE (He et al., 2022), PvT (Wang et al., 2022a), SEGFORMER (Xie et al., 2021), SWINT (Liu et al., 2021)) to enhance RGB feature expressiveness through ensemble learning. EITLNET (Guo et al., 2024), although built on SegFormer and primarily driven by noiseprint signals, incorporates RGB inputs to support spatial localization and reasoning. Similarly, ERMPC (Li et al., 2023a) adopts a parallel dual-stream design using ResNet50 to process both RGB and noiseprint features, demonstrating how multi-branch architectures can exploit RGB information even under compression artifacts. MUN (Liu et al., 2025) further advances dual-stream RGB-noise fusion by employing parallel ConvNeXt V2 (Woo et al., 2023) backbones and introducing a Multi-scale Max-pooling Query (MMQ) module that enables cross-domain feature correlation, allowing RGB features to guide the extraction of manipulation-relevant noise patterns. This architectural diversity reflects the adaptability of RGB features across detection scenarios and technical pipelines.

## 5.2 Frequency Features

Frequency features convert images from the spatial domain into the frequency domain by applying transformations such as the Fast Fourier Transform (FFT) or Discrete Cosine Transform (DCT). This transformation enables the detection of periodic structures and spectral inconsistencies that are often imperceptible in the RGB space. Previous research has demonstrated that manipulations such as splicing, copy-move, inpainting, and AI-generated synthesis frequently disturb the natural frequency distributions found in authentic images. As a result, frequency-based analysis is particularly effective in identifying artifacts introduced by JPEG compression, resampling operations, and the characteristic spectral signatures produced by generative models. These signals are often subtle and not easily captured by RGB-based methods, which primarily rely on spatial information. The principal advantage of frequency features lies in their ability to detect statistical anomalies and periodic patterns that remain invisible in the spatial domain, making them especially useful for the detection of high-quality or visually coherent forgeries. However, frequency features are also sensitive to preprocessing operations such as filtering and scaling.

To fully leverage spectral information for forgery detection, recent methods have proposed a variety of architectural designs that integrate frequency-domain features. DA-HFNet (Liu et al., 2024d) incorporates a dedicated frequency extraction branch within an HRNet (Sun et al., 2019) backbone, enabling the model to preserve high-resolution spectral details while detecting inconsistencies across different generative sources. This design facilitates the fusion of frequency and spatial features, improving the model's ability to identify subtle manipulations. Mesorch (Zhu et al., 2025) adopts a dual-path ensemble architecture that combines a ResNet50-based CNN stream with multiple Transformer variants including MAE, PvT, Seg-Former, and SwinT. This setup decouples the processing of high- and low-frequency components, allowing the CNN branch to focus on local spectral anomalies while the Transformer branch models global semantic relationships, achieving robust detection across manipulation types. HiFi-Net (Guo et al., 2023) and its extension HiFi-Net++ (Guo et al., 2025c) are both built upon ResNet50 and Feature Pyramid Network (FPN) backbones. These models apply Laplacian of Gaussian (LoG) filters to the intermediate CNN feature maps, explicitly extracting frequency-sensitive patterns. By employing multi-scale branches, they enhance the model's ability to perform fine-grained classification and precise localization of manipulation artifacts. HiFi-Net++ (Guo et al., 2025c) further extends this framework to the multimodal setting by incorporating Transformer-based modules that process textual information, illustrating the compatibility of frequency features with cross-modal forgery analysis. MSCCNet (Miao et al., 2023), another frequency-driven approach based on ResNet50, introduces a multi-spectral class center module that utilizes DCT to decompose semantic features into distinct frequency bands. This architecture is specifically designed to extract class-specific high-frequency cues associated with different forgery types, and is trained using a supervised strategy tailored to enhance the discriminative power of frequency features. ESIDE (Wu et al., 2025) leverages frequency-domain discrepancies between real and synthetic images by perturbing inputs across diffusion timesteps, revealing distinctive high-frequency patterns in the Fourier spectrum for detection without requiring task-specific annotations. In summary, frequency features provide a complementary perspective to spatial-domain analysis and have been successfully integrated into a range of architectures. Their capacity to capture imperceptible manipulation signals makes them a powerful tool in both supervised and unsupervised detection pipelines, particularly when paired with multi-branch or attention-based designs that accommodate the complexity of spectral patterns.

## 5.3 Noiseprint Features

Noiseprint features represent a distinct class of forensic cues that capture content-independent statistical patterns arising from underlying imaging processes, including sensor response, compression encoding, and image reconstruction. Unlike RGB features that emphasize spatial structure and color composition, noiseprint features are extracted through high-pass filtering or residual modeling, with the goal of isolating high-frequency signal components. These features are particularly effective at revealing subtle perturbations introduced by manipulations, such as edge smoothing, compression artifacts, and inpainting operations, which are often imperceptible in other feature domains. In addition to general residual signals, a related form known as cam-based noise is also relevant in forgery detection. This category includes sensor-specific traces such as photo-response non-uniformity (PRNU), lens distortions, thermal noise, and artifacts introduced by the

image signal processing pipeline. These characteristics are inherently embedded in images captured by real cameras. In contrast, AI-generated content typically lacks such acquisition-specific patterns, making the presence or absence of cam-based noise a useful indicator of authenticity. Together, these signal-level residuals provide complementary robustness for detecting subtle manipulations, particularly in high-quality or heavily compressed forgeries where pixel-level or semantic features may be insufficient.

Several models have been proposed to effectively extract and use noiseprint features. TRUFOR (Guillaro et al., 2023) combines a lightweight DnCNN backbone with a SegFormer architecture. It uses a pretrained residual extractor to generate noise maps that suppress image content while preserving manipulation cues, and applies dual attention modules to improve focus on suspicious regions. A self-supervised training strategy enables the model to work without dense annotations. ERMPC (Li et al., 2023a) adopts a two-stream ResNet50 design to process RGB and noise residuals in parallel, using patch-level supervision and multi-scale fusion to improve detection under compression. Similarly, MUN (Liu et al., 2025) employs dual ConvNeXt V2 encoders for parallel RGB and Noiseprint++ extraction, with cross-domain feature correlation enabling more effective noise pattern discovery. EITLNET (Guo et al., 2024) extends SegFormer by integrating handcrafted noise maps and semantic features using a cross-scale attention mechanism, enhancing its ability to capture diverse forgery patterns. *Bi et al.* propose a self-supervised method that learns to detect forgery boundaries by modeling JPEG compression differences, reducing reliance on detailed annotations. DA-HFNET (Liu et al., 2024d) includes a dedicated noise branch alongside RGB and frequency branches in an HRNet framework, using spatial attention to guide the model toward manipulation-sensitive regions. *Zhu et al.* (Zhu et al., 2024) further introduce a noise-guided attention mechanism that applies pseudo-perturbations to residual signals, helping the model focus on vulnerable areas. Overall, noiseprint-based methods are particularly useful in identifying tampering traces that are not easily captured by other features. Their ability to expose low-level inconsistencies makes them a valuable component of explainable forgery detection systems.

## 5.4 Representation Features

Representation features denote high-level semantic abstractions acquired automatically through deep neural networks, with Transformer-based architectures playing a central role. Unlike handcrafted low-level features such as RGB, frequency, or noiseprint signals, these features are learned end-to-end and encode complex patterns encompassing object semantics, spatial dependencies, and cross-modal associations. Such features enhance model generalization across varied manipulation types while supporting explainability through semantic masks and textual justifications. In multi-modal forgery detection, representation learning is particularly critical, as it enables effective alignment across heterogeneous modalities—including visual, textual, and auditory signals. Recent methods commonly integrate large-scale pre-trained models such as ViT, CLIP, and LLaVA to construct a unified embedding space that jointly encodes multi-modal inputs, thereby improving both generalization and explainability.

In single-modal forgery detection, diverse network architectures are tailored to extract discriminative representations from specific data types. CSR-NET (Zhang et al., 2024b) introduces a multi-scale semantic extraction module supervised by spline regression loss for refined mask prediction. GIFL (Zhao et al., 2025) enhances a ViT-L/14 encoder with deep residual blocks to improve tampering sensitivity. DELOCATE (Hu et al., 2024) adopts a dual-branch framework combining frame-level ViT and ResNet-18 to capture spatiotemporal cues in videos. ADAIFL (Li et al., 2024) proposes an adaptive cross-scale reconstruction module fusing RGB and ViT features for improved domain generalization. MESORCH (Zhu et al., 2025) leverages CNN backbones and diverse Transformer decoders (e.g., MAE, SwinT) to disentangle frequency streams and jointly model local and global contexts. *Khormali et al.* (Khormali & Yuan, 2024) employ masked image modeling and graph convolutions in a self-supervised setup to extract structured facial representations. OMNIGUARD (Zhang et al., 2024c) uses a VAE to generate forgery prior maps, which are fused with ViT outputs to refine attention on suspicious areas. SAFIRE (Kwon et al., 2025) integrates SAM-based segmentation with ViT to produce attention-guided features. *Hong et al.* (Hong et al., 2024) apply contrastive learning to refine ViT-based localization, while DIFFFORENSICS (Yu et al., 2024b) injects diffusion priors into ViT encoders to generate more discriminative features.

Building upon these single-modal advances, multi-modal models further extend representation learning by establishing shared embedding spaces across modalities. DE-FAKE (Sha et al., 2023) introduces a visual-

textual consistency mechanism using ViT-B/32 and MLP encoders. HAMMER (Shao et al., 2023) fuses ViT-B/16 and BERT embeddings via hierarchical attention. MGCA (Guo et al., 2025a) designs multi-granularity alignment blocks between ViT and Vicuna features. FFAA (Huang et al., 2024a) aligns ViT-L/14 and T5 features through LLaVA-v1.6-7B. FORGERYGPT (Liu et al., 2024c) and FORGERYTALKER (Lian et al., 2024) combine ViT and Vicuna with Q-Former modules for joint classification and explanation. M2F2-DET (Guo et al., 2025d) proposes a lightweight fusion of EfficientNet-B4, ViT-L/14, and Vicuna. FAKESHIELD (Xu et al., 2024) and LEGION (Kang et al., 2025) stack ViT, SAM, and language models (e.g., LLaVA, GLaMM) for vision-language fusion. SIDA (Huang et al., 2024b) uses Lang-SAM and LISA for language-conditioned masking. FORGERYSLEUTH (Sun et al., 2024) aligns ViT-H masks with LLaVA-generated captions through a consistency loss. TRUTHLENS$^{\text{GOOGLE}}$ (Kundu et al., 2025a) jointly trains SigLIP-So400m/14 and DINOv2 with Gemma2 and PaliGemma2 for vision-language space unification. HIFI-NET++ (Guo et al., 2025c) integrates frequency pyramids with language embeddings. ESIDE (Wu et al., 2025) perturbs DDIM latents to produce frequency-aware masks, fused with GPT-4o features. EDITSCOUT (Nguyen et al., 2024) promotes cross-modal consistency between diffusion-edited images and text. *Yu et al.* (Yu et al., 2025b) and DIMODIF (Koutlis & Papadopoulos, 2024) project audio and visual features into shared spaces using ImageBind-Huge or Transformer-based ASR/VSR fusion to capture modality inconsistencies. Recent MLLM-based methods further advance explainable detection by leveraging large-scale vision-language models. LVLMDFD (Yu et al., 2025a) fine-tunes pre-trained MLLMs to generate textual explanations for deepfake detection. AIGI-HOLMES (Zhou et al., 2025) augments LLaVA's encoder with a noiseprint-based visual expert and employs direct preference optimization to align model outputs with human judgment. *Song et al.* (Song et al., 2025) introduce a multi-paradigm prompting strategy across various MLLMs to enhance reasoning consistency. RLSBI (Jiang et al., 2026) adapts LLaVA-1.5 (Liu et al., 2023) for deepfake detection through specialized prompt engineering and semantic-guided localization. Collectively, these methods underscore the pivotal role of representation learning in advancing both single- and multi-modal forgery detection paradigms.

## 5.5 The Role of Features in Explainability

To understand how different feature types serve the three explainability dimensions introduced in Section 4.3, we analyze the mapping between feature-driven strategies and explainability tasks across the methods in Table 3. The results are summarized in Table 4. Several patterns emerge from this analysis:

*Observation 1: Low-level features concentrate on Forgery Localization.* As Table 4 shows, every RGB, frequency, and noiseprint method in our survey supports FL, yet their engagement with FA and FJB remains sparse. This likely reflects the inherently spatial nature of these features: RGB features retain pixel-level structure suited for mask generation, frequency features capture localized spectral inconsistencies such as JPEG grid misalignment that can be mapped to tampered regions, and noiseprint features produce residual maps encoding manipulation traces at specific locations. The limited presence in FJB may stem from the fact that low-level cues can indicate *where* anomalies exist but lack the semantic capacity to explain *why* an image appears manipulated.

*Observation 2: Frequency features show underexploited potential for Forgery Attribution.* Among the frequency-based methods, only HIFI-NET Guo et al. (2023) and HIFI-NET++ Guo et al. (2025c) leverage spectral fingerprints for attribution, exploiting the tendency of different generators to leave distinct frequency signatures—for instance, GAN upsampling artifacts versus diffusion-specific noise distributions. The scarcity of methods pursuing this direction suggests that spectral attribution remains an open opportunity. One barrier may be that frequency evidence, while informative for automated classification, is abstract and difficult to communicate to human users, which could also explain its minimal adoption for FJB.

*Observation 3: Noiseprint features have untapped potential beyond localization.* Notably, every noiseprint method in our survey targets FL exclusively. This is somewhat surprising given that camera-specific patterns such as PRNU could in principle support FA by distinguishing real captures from generated content, and that noise inconsistency could serve as forensic evidence for FJB. The absence of such applications likely reflects practical challenges: building noise fingerprint databases for attribution or integrating language generation modules for explanation requires engineering efforts that existing noiseprint pipelines have not yet undertaken.

Table 4: Mapping between feature-driven strategies and explainability tasks, based on the methods surveyed in Table 3.

| Feature Type | #Methods | FL | FA | FJB | Primary Explainability Role |
|---|---|---|---|---|---|
| RGB | 11 | 11 (100%) | 2 (18%) | 0 (0%) | Spatial localization via pixel-level mask generation |
| Frequency | 9 | 9 (100%) | 2 (22%) | 1 (11%) | Localization and limited model fingerprint attribution |
| Noiseprint | 7 | 7 (100%) | 0 (0%) | 0 (0%) | Localization via noise residual mapping |
| Representation | 32 | 26 (81%) | 11 (34%) | 16 (50%) | Full-spectrum: localization, attribution, and judgment reasoning |

*Observation 4: Representation features dominate Forgery Judgment Basis.* Perhaps the most striking pattern in Table 4 is that roughly half of the representation-based methods support FJB, while virtually none of the methods relying solely on other feature types do so. A plausible explanation is that representation features, learned through vision-language models such as LLAVA (Liu et al., 2023), GPT-4 (Achiam et al., 2023), and BLIP-2 (Li et al., 2023b), encode semantic abstractions that can be decoded into natural language—a capability essential for articulating *why* a manipulation is detected. Representation features also exhibit the broadest coverage across all three dimensions, reflecting the flexibility of learned embeddings for diverse tasks: from mask generation in FAKESHIELD (Xu et al., 2024) and LEGION (Kang et al., 2025), to cross-modal mismatch detection in HAMMER (Shao et al., 2023), to textual explanation generation in FORGERYSLEUTH (Sun et al., 2024) and FORGERYGPT (Liu et al., 2024c).

The complementary distribution observed in Table 4, where low-level features excel at FL while representation features are needed for FJB, helps explain the prevalence of multi-stream architectures such as HIFI-NET++ (Guo et al., 2025c), DA-HFNET (Liu et al., 2024d), and MESORCH (Zhu et al., 2025). This suggests that comprehensive explainability may require architectures that explicitly bridge low-level forensic cues with high-level semantic reasoning, for example through modules that translate spatial or spectral evidence into natural language justifications.

## 6 Benchmark Datasets and Evaluations

This section examines representative benchmark datasets and evaluation protocols supporting various forgery detection tasks. Commonly used datasets are first categorized by modality and construction strategy in Table 5. The discussion then turns to evaluation metrics for classification, localization, and judgment basis tasks, emphasizing their importance in assessing model performance across different explainability dimensions.

### 6.1 Datasets

Benchmark datasets are essential to forgery detection, enabling not only standard binary classification but also advanced tasks such as localization, attribution, and explanation. Most existing datasets are constructed using generative models such as GANs or diffusion-based methods to synthesize controlled forgeries that follow consistent patterns (Rössler et al., 2019; Dolhansky et al., 2020; Li et al., 2020). A smaller subset of benchmarks is curated from social media platforms or user-submitted content, where manipulations often appear as mismatched image and text pairs or unstructured edits (Miao et al., 2024; Wu et al., 2025). These real-world forgeries are generally more diverse and less predictable, presenting a higher degree of challenge due to their contextual variability and the lack of standardized annotation.

To reflect the distinctions in modality and supervision level, we divide the datasets into two categories: public source datasets and curated datasets introduced in recent methods. Among the 48 methods we surveyed, 22 introduced their own curated datasets. To better understand how these datasets were constructed, we also tracked their source datasets—*i.e.,* the underlying datasets used to generate forgeries. Importantly, the `"Tasks"` column in Table 5 aggregates all downstream tasks supported by each curated dataset across the methods that used them, which implies the detection capabilities those datasets are designed to evaluate. Based on modality, the datasets broadly fall into two groups: single-modal vision datasets and multi-modal vision-text datasets, each supporting different aspects of explainable detection.

Table 5: Overview of **public** and **curated** datasets commonly used in forgery detection.

| Dataset | Modality | Tasks† | Fake # w/ Type | Fake Source | Real # w/ Type | Real Source | Annotator | Availability | Remark |
|---|---|---|---|---|---|---|---|---|---|
| **Public Source Dataset** | | | | | | | | | |
| TGIF (Mareen et al., 2024) | Image | FL | 75K Images | SD2, SDXL, Adobe Firefly | 3.1K Images | MS-COCO | - | ✓ | Text-guided inpainting forgery dataset with pixel-level masks; includes spliced and fully regenerated versions. |
| SynthBuster (Bammey, 2023) | Image | FL | 9K Images | DMs | 9.6K Images | RAISE Dang-Nguyen et al. (2015), Dresden Gloe & Böhme (2010) | - | ✓ | Fully AI-generated images from 9 diffusion models; designed for synthetic image detection. |
| SIDBench (Schinas & Papadopoulos, 2024) | Image | FL | - | GANs, DMs | - | Laion Schuhmann et al. (2022), Raise Dang-Nguyen et al. (2015) | - | ✓ | A benchmarking framework integrating 11 SID models and multiple datasets for standardized evaluation. |
| FF++ (Rössler et al., 2019) | Image | FL, FJB, FA | 1.8M Facial Frames | DeepFakes, Face2Face, FaceSwap, NeuralTextures | 509K Facial Frames | YouTube | - | ✓ | Real (from 1K videos) and fake facial images (from 4K videos) paired with pixel-level forgery masks. |
| WildDeepfake (Zi et al., 2020) | Image | FL | 3.5K Facial Sequences | Internet | 3.8K Facial Sequences | Internet | Human | ✓ | 1M face images from 7K face sequences extracted from 707 internet videos. |
| Celeb-DF (V2) (Li et al., 2020) | Image | FL | 21M Facial Frames | Faceswap | 2M Facial Frames | Youtube | - | ✓ | 590 real and 5,639 fake videos of celebrities generated by DeepFake synthesis pipeline. |
| CASIAv2 (Dong et al., 2013a) | Image | FL | 5K Images | Splicing, Copy-Move, Removal | 7K Images | - | - | ✓ | Images with mask pairs. |
| ALASKA (Ruiz et al., 2021) | Image | FL | 2M Images | Generated with J-UNIWARD | 2M Images | ALASKA#2, BOSS, Dresden, RAISE, StegoApp, Wesaturate | - | ✓ | Real and Fake (quality factor 75) created from RAW public datasets using J-UNIWARD. |
| Fantasitic-Reality (Kniaz et al., 2019a) | Image | FL | 16K Images | Splicing with Generative Retoucher | 16k Images | - | Human | ✓ | Real and spliced images with pixel-level tamper masks, instance masks, and 10-class semantic labels including person, car, bus, etc. |
| Protocol-CAT (Ma et al., 2024) | Image | FL | - | CASIA (Dong et al., 2013a), Coverage, Columbia, NIST16, IMD2020, NC2016, NC2017, Defacto | - | CASIA v2.0, Coverage, Columbia, NIST16, IMD2020, NC2016, NC2017, Defacto | - | ✓ | This provided through a codebase referenced IMDL-BenCo. |
| Synthesized Dataset (Liu et al., 2025) | Image | FL | 156K Images | NIST16, CASIA, IMD2020, CocoGlide, Wild, CASIA, ADE20K | - | - | - | ✓ | There are 156006 synthesized images(140432 for training, 7787 for validation, and 7787 for testing). |
| WildFake (Hong & Zhang, 2024) | Image | FJB | 2M Images | GANs, DMs, AEs | 1M Images | LAION-5B, Wukong, COCO, ImageNet, FFHQ, CelebA-HQ, AFHQ, LSUN | - | ✓ | WildFake contains 3.69M images (1M real, 2M fake) |
| DF40 (Yan et al., 2024b) | Image | FJB | 1M Images | PixArt-α, Face-swapping, Face-reenactment, Entire Face Synthesis, Face Editing | - | - | - | ✓ | Large-scale deepfake benchmark with 40 manipulation methods. |
| DeepfakeBench (Yan et al., 2023) | Video | FL, FJB | 133k Videos | FF++, CelebDF-v1, CelebDF-v2, DFD, DFDC, UADFV, FaceShifter, DeeperForensics-1.0 | 78K Videos | FF++, CelebDF-v1, CelebDF-v2, DFD, DFDC, UADFV, FaceShifter, DeeperForensics-1.0 | - | ✓ | DeepfakeBench provides preprocessed facial deepfake datasets from multiple public sources, including cropped faces, masks, and landmarks, enabling direct image- and video-level evaluation. |
| DeeperForensics-1.0 (Jiang et al., 2020) | Video | FL | 5M Facial Frames | Faceswap (Tampered via DF-VAE) | 12.6M Frames | From 100 Paid Actors | Human | ✓ | Frames from 60K facial videos with diverse perturbations. |
| DFDC (Dolhansky et al., 2020) | Video | FL | 100K Video Clips | AEs, GANs, TTS Audio Swap | 28K Video Clips | 3,426 Paid Actors | - | ✓ | Videos with face/audio swaps, from 3,426 paid actors in diverse lighting and scene conditions. |
| DD-VQA (Zhang et al., 2024d) | Image-Text | FL, FJB | 3K Facial Images | FF++ | 1k Videos | FF++ | Human | ✓ | Facial images and 14K Q-A pairs. |
| MagicBrush (Zhang et al., 2023) | Image-Text | FL | 10K Images | DALL·E 2 | 5K real source images | MSCOCO | Human | ✓ | Real and Fake with multi-turn instruction-guided edits, covering object addition, removal, attribute change, etc. |
| AutoSplice (Jia et al., 2023) | Image-Text | FL | 3.5K Images | Visual News with DALL·E 2 | 2K Images | Visual News | Human | ✓ | Fake and Real images, each with pixel-level forgery mask, based on news photos and text edits from real media. |
| DVF (Song et al., 2024) | Image-Text | FL, FJB | 4K Videos | TikTok, OpenSora, VideoCrafter1 | 2.7K Videos | Internvid-10M, YouTube-8M | - | ✓ | AI-generated videos and real-world videos covering diverse semantics and video qualities. |
| FakeAVCeleb (Khalid et al., 2021) | Video-Text | FL | 19.5K Videos | Faceswap on VoxCeleb2 | 500 Videos | VoxCeleb2 | - | ✓ | Videos with all 4 combinations of real/fake video and audio (ARVR, AFVR, ARVF, AFVF) with lip-sync alignment, diverse ethnicity, and gender. |
| LAV-DF (Cai et al., 2022) | Video-Text | FL | 99K Videos | VoxCeleb2 | 36K Videos | VoxCeleb2 | - | ✓ | Videos from 153 identities; includes real and multimodally manipulated clips (audio-only, video-only, both), with frame-level sentiment and timing labels. |
| AVDeepfake1M (Cai et al., 2024) | Video-Text | FL | 860K Videos | ChatGPT | 286K Videos | VoxCeleb2 | - | ✓ | Videos from 2K subjects, with insert/delete/replace edits across audio, video, or both; include 1.8K hours of high-quality AV data. |
| **Curated Dataset** | | | | | | | | | |
| OFV2 (Miao et al., 2024) | Image | FL | 70K Facial Images | OpenForensics (Tampered via GANs) | - | Open Images (Kuznetsova et al., 2020) | - | ✗ | A multi-face forgery image dataset with forged images and pixel-level forgery masks. |
| Miao et al., (Miao et al., 2024) | Image | FL | 20K Facial Videos | FFIW (Multiface manipulations only) | - | FFIW | - | ✗ | Multi-face forgery video frames(Faceswap,Fsgan,Deepfacelab) with masks. |
| Miao et al., (Miao et al., 2023) | Image | FL | 273K Facial Frames | P-FF++ (Rossler et al., 2019), Dolos (Tăntaru et al., 2024) (DMs, GANs, LaMa) | 188K Frames | P-FF++ (Rossler et al., 2019), Dolos (Tăntaru et al., 2024) | - | ✓ | Real-Fake facial image pairs with masks. |
| DA-HFNet (Liu et al., 2024d) | Image | FL | 21K Images | GANs, DMs, Inpaint Anything, StyleCLIP | 20K Images | COCO2017 (Lin et al., 2014), ImageNet (Deng et al., 2009) | - | ✗ | Combining full and partial image forgeries generated by GAN and Diffusion models guided by text or image inputs. |
| HiFi-IFDL (Guo et al., 2025c) | Image | FA, FL | 1.3M Images | LSUN, CelebAHQ, FFHQ, AFHQ, Youtube (Tampered via GANs, DMs, Faceshifter) | - | - | - | ✓ | Fake images are paired with high-resolution binary forgery masks. |
| SafireMSExpert (Kwon et al., 2025) | Image | FA, FL | 238 Images | DPReview, COCO2017 | - | - | Human | ✗ | A high-quality, manually constructed image forgery dataset with multi-source edits and pixel-level masks designed by human experts. |
| SafireAuto (Kwon et al., 2025) | Image | FA, FL | 123K Images | DPReview, COCO2017 (Tampered via Copy-Move, Splicing, Removal, Generative Reconstruction) | 30K Images | DPReview, COCO2017 | - | ✗ | A large-scale automatically generated dataset using SAM-based masks over real images from COCO and DPReview, with 4 editing types and paired masks. |
| X-FAS (Zhang et al., 2024a) | Image | FA, FL | 56 Spoof Images | CASIA-FASD | - | - | Human | ✗ | 56 CASIA-FASD spoof images annotated by experts with four semantic-level spoofing evidences for evaluating explainable face anti-spoofing methods. |
| GENHARD (Wu et al., 2025) | Image | FA | 108K Images | GenImage (Zhu et al., 2023) | 112K Images | GenImage (Zhu et al., 2023) | - | ✗ | A curated set of "hard" real and fake images from GenImage, chosen based on consistent misclassification. |
| ForgeryADE (Zhao et al., 2025) | Image | FA, FL | 177K Images | ADE 20 (Zhou et al., 2017) | 22K Images | ADE20K | - | ✗ | A dataset containing manipulated images and authentic images, with diverse occlusion masks and 8 modern forgery methods. |
| Manual-Fake (Miao et al., 2024) | Video | FL | 1K Videos | YouTube, TikTok, Bilibili | 1K Videos | Internet | - | ✗ | A multi-face forgery video dataset for evaluation on real-world deepfakes, with pixel-level annotations |
| Sha et al., (Sha et al., 2023) | Image-Text | FA | 40K Images | DMs | 40K Images | MSCOCO (Lin et al., 2014), Flickr30k (Young et al., 2014) | - | ✗ | Images generated from textual prompts using DMs. |
| DGM⁴ (Shao et al., 2023) | Image-Text | FA, FL | 152K Image-Text Pairs | CelebA-HQ, VisualNews (Liu et al., 2020) | 77K Image-Text Pairs | VisualNews | - | ✓ | Image-text news-style pairs with manipulated pairs covering face and text edits, annotated for multimodal detection and grounding. |
| MMTD-Set (Xu et al., 2024) | Image-Text | FA, FL, FJB | 51k Images | Fantastic Reality (Kniaz et al., 2019b), CASIAv2 (Dong et al., 2013b), FaceAPP (Stehouwer et al., 2019), AIGC-Editing (Lugmayr et al., 2022) | 54k Images | Fantastic Reality (Kniaz et al., 2019b), CASIAv2 (Dong et al., 2013b), FFHQ (Karras et al., 2019), COCO (Lin et al., 2014) | GPT-4o | ✓ | General image and tampered masks, with instruction. |
| FFA-VQA (Huang et al., 2024a) | Image-Text | FA, FL, FJB | 67K Frames | FF++ (Rössler et al., 2019), DFFD, GanDiffFace | 27K Frames | FF++ (Rössler et al., 2019), Celeb-DF-V2 (Li et al., 2020), DFFD (Stehouwer et al., 2019) | GPT-4o, Human | ✓ | Face images with identity swap, attribute manipulation, or entire face synthesis, each paired with GPT-generated VQA analysis. |
| ForgeryAnalysis (Sun et al., 2024) | Image-Text | FL, FJB | 50K Image-Text Pairs | MIML (Qu et al., 2024), CASIA2 (Dong et al., 2013b), DEFACTO (Mahfoudi et al., 2019), AutoSplice (Jia et al., 2023) | - | - | Human | ✓ | 50k manipulated image-mask-analysis text pairs. |
| AMG (Guo et al., 2025a) | Image-Text | FA | 2K Posts | Instagram, Facebook, Twitter | 3K Posts | Fact-checking sites, Reuters (Reuters, 2023), NewsNation (NewsNation, 2023) | Human | ✓ | Social media posts from 2016-2024, annotated with fine-grained attribution reasons for misinformation. |
| MMTT (Lian et al., 2024) | Image-Text | FA, FL, FJB | 128K Facial Images | CelebAMaskHQ (Zhu et al., 2022), FFHQ (Karras et al., 2019) (Tampered via GANs (Goodfellow et al., 2014), Transformers (Vaswani et al., 2017a), DMs (Ho et al., 2020b)) | 100K Facial Images | CelebAMask-HQ, FFHQ | Human | ✗ | Each of the face images is paired with a high resolution binary forgery mask and a detailed human written caption describing manipulated regions and their artifacts. |
| Liu et al., (Liu et al., 2024c) | Image-Text | FA, FL, FJB | 768K Image-Caption Pairs | MSCOCO (Lin et al., 2014) (Tampered via Splicing, Copy-Move, Removal) | - | - | GPT-4 | ✗ | A multi-granularity dataset containing image-caption pairs generated using traditional manipulate techniques. |
| GENEXPLAIN (Wu et al., 2025) | Image-Text | FA, FJB | 54K Images | GenImage (Zhu et al., 2023) | - | - | GPT-4o | ✗ | A high-resolution Image-Text dataset with 14 flaw categories, GPT-4o-generated and refined explanations for each fake image. |
| SID-Set (Huang et al., 2024b) | Image-Text | FA, FL, FJB | 200K Images | FLUX (Flux Model Team, 2024), KANDINSKY3.0 (Arkhipkin et al., 2023), SDXL (Podell et al., 2023), AbsoluteReality (Absolutereality Model Team, 2024), LD (CompVis, 2023) | 100K Images | OpenImages V7 (Google Research, 2022) | GPT-4o, Human | ✓ | A dataset for deepfake detection on social media, includes segmentation masks and 3K textual explanations. |
| SynthScars (Kang et al., 2025) | Image-Text | FA, FL, FJB | 12K Images | RichHF-18K (Liang et al., 2024), Chameleon (Yan et al., 2024a), FFAA (Huang et al., 2024a) | - | - | Human | ✓ | A dataset containing images with irregular polygon masks, detailed explanations, and classification into 3 artifact types: physics, distortion, and structure. |
| Holmes-Set (Zhou et al., 2025) | Image-Text | FJB | 65K Images-Text Pairs | CNNDetection (Wang et al., 2020), GenImage (Zhu et al., 2023) | - | - | Human | ✓ | Holmes-SFTSet/DPOSet: Explanation-rich SFT and human-preference datasets enabling aligned explainable AIGI detection. |

†: **FL**: Forgery Localization, **FA**: Forgery Attribution, **FJB**: Forgery Judgment Basis.

## 6.2 Single-Modal Vision Datasets

**Image-Modal Datasets.** Single-modality visual datasets have underpinned the rapid progress of image and video forgery detection. These datasets typically consist of real and forged visual content and are

primarily designed for facial forgery detection, object-level splicing, or frame-wise tampering. Early datasets such as FF++ (Rössler et al., 2019) and CELEB-DF (V2) (Li et al., 2020) provided large-scale facial manipulations and played a foundational role in training supervised detectors. Most image-based datasets only contain visual samples with binary or pixel-level labels, thus they are predominantly used for forgery localization. Several datasets focus specifically on facial manipulations, such as FF++ (Rössler et al., 2019), WILDDEEPFAKE (Zi et al., 2020), and CELEB-DF (V2) (Li et al., 2020), where synthetic faces are generated using DeepFakes, Face2Face, FaceSwap, or NeuralTextures applied to real face data. In addition, hybrid datasets that combine facial and natural images, including TGIF Mareen et al. (2024), SYNTHBUSTER Bammey (2023), SIDBENCH Schinas & Papadopoulos (2024), CASIAv2 (Dong et al., 2013a), ALASKA (Ruiz et al., 2021), GENHARD (Wu et al., 2025), WILDFAKE (Hong & Zhang, 2024), and others (Kniaz et al., 2019b; Ma et al., 2024; Liu et al., 2024d; Guo et al., 2025c; Kwon et al., 2025; Zhang et al., 2024a; Zhao et al., 2025; Yan et al., 2024b) are constructed to support evaluation across more diverse manipulation scenarios.

These datasets are typically constructed from diverse sources and include various forgery methods, ranging from traditional splicing and removal to advanced model-based generation (*e.g.,* text-to-image editing, repainting, inpainting via diffusion models). While most of these datasets are limited to image–mask pairs and thus suitable for localization tasks, it is worth noting that some works (Yu et al., 2025b; Khormali & Yuan, 2024; Hong et al., 2024; Hu et al., 2024) extend the use of source datasets like FF++ (Rössler et al., 2019) to also support attribution and explanation tasks via additional annotations or processing.

**Video-Modal Datasets.** In contrast to image forgeries, relatively few works in our survey focus on video-level forgery detection. Among them, DeeperForensics-1.0, DFDC, DeepfakeBench and MANUAL-FAKE (Jiang et al., 2020; Dolhansky et al., 2020; Yan et al., 2023; Miao et al., 2024) represent three distinct strategies for constructing annotated video datasets, each supporting forgery localization, robustness evaluation, or generalization studies. DeeperForensics-1.0 is built from 100 paid actors and generates over 5 million forged frames using DF-VAE (Jiang et al., 2020), with simulated perturbations to assess robustness to degradations (Jiang et al., 2020). DFDC remains the largest competition-scale benchmark for deepfake detection, featuring over 3,000 participants and a wide variety of manipulation types (GAN, AutoEncoder, audio swap), making it ideal for large-scale multimodal forgery analysis (Dolhansky et al., 2020). In contrast, Manual-Fake is derived from social platforms (YouTube, TikTok, Bilibili), capturing naturally occurring manipulations with pixel-level annotations (Miao et al., 2024). Although smaller in scale, it provides high authenticity and multi-face occlusion scenarios, posing unique challenges. Compared to image datasets, these video benchmarks offer greater complexity in temporal modeling, scene variation, and multimodal integration, making them especially valuable for advancing explainable video forgery detection.

### 6.3 Vision–Text Multimodal Datasets

**Image–Text Multimodality.** In image–text datasets for forgery detection, the text component can take multiple forms depending on the dataset's design. One common format is a descriptive caption summarizing the visual content, typically describing the scene or objects within the image. Such captions provide semantic context and are used to assess cross-modal consistency. For example, they help ensure alignment between the image and its textual description, or improve model performance through cross-domain feature fusion and alignment (Sha et al., 2023). Another format involves explanatory annotations that specify manipulation-related information, such as the type of operation (*e.g.,* splicing, copy-move, or AI generation), the exact tampered regions, or the reasoning beyond the real/fake classification (Shao et al., 2023; Guo et al., 2025a). These annotations provide explicit forensic cues and help models learn to associate anomalies in the image with specific features or regions, thereby enhancing explainability. Some datasets, such as DD-VQA and FFA-VQA, adopt a question–answer format, where each image is paired with targeted questions about its authenticity or content. The corresponding answers include both the classification result and a supporting explanation. This QA format encourages stepwise reasoning and typically requires the model to express its decision-making process in natural language. Lastly, many datasets generate explanation texts during the tampering process, either through manual annotation or by using prompt-based methods with large language models (Achiam et al., 2023; Liu et al., 2023; Zhou et al., 2025). These various text forms reflect different supervisory signals and learning objectives: caption-based datasets emphasize cross-modal consis-

tency between images and descriptions; explanatory annotations support visual attribution; and QA-style supervision promotes interactive and explainable reasoning. Training strategies should therefore be selected in accordance with the target task and the characteristics of the dataset.

**Video–Text Multimodality.** Several recent benchmarks have been introduced to support multimodal forgery detection involving both video and text or audio. These datasets consist of paired video and audio content, where either modality may be real or manipulated. FAKEAVCELEB (Khalid et al., 2021) contains 19.5K video samples generated by applying FaceSwap to VOXCELEB2 (Chung et al., 2018). The dataset includes four combinations of authenticity across the audio and video streams: real video with real audio (ARVR), real video with fake audio (ARVF), fake video with real audio (AFVR), and fake video with fake audio (AFVF). All samples are lip-synced and include a diverse range of demographic attributes such as ethnicity and gender. LAV-DF (Cai et al., 2022) comprises 99K videos across 153 unique identities, constructed from VoxCeleb2 clips. It contains real as well as manipulated samples, with forgery applied to the audio only, video only, or both. Each clip is annotated with frame-level sentiment labels and temporal alignment information. AVDEEPFAKE1M (Cai et al., 2024) consists of 860K synthetic video clips created using ChatGPT for dialogue generation and VoxCeleb2 as the visual source. The manipulations include insertion, deletion, and replacement in the audio, video, or both. The dataset comprises approximately 1.8K hours of video and audio content and features 2K distinct speaker identities.

## 6.4   Evaluation Metrics

We categorize evaluation metrics in forgery detection into three main groups: classification-oriented, localization-oriented, and explanation-oriented. These groups correspond to different aspects of the explainability framework developed in this survey. Classification-oriented metrics primarily assess the binary or multi-class detection performance that underlies all three dimensions (FL, FA, and FJB). Localization-oriented metrics directly evaluate FL by measuring the spatial or temporal precision of identified manipulated regions. Explanation-oriented metrics are most closely tied to FJB, assessing whether generated textual rationales faithfully and coherently convey the reasoning behind detection decisions; they also partially evaluate FL when explanations include spatial references, and FA when attribution reasoning is assessed. Classification and localization are well-defined tasks with widely adopted quantitative metrics. In comparison, textual explanation is an open-ended objective that often requires more flexible and interpretive forms of evaluation, particularly in scenarios involving Multimodal Large Language Models (MLLMs). This section provides an overview of these metric categories and discusses how they relate to the evaluation of explainable detection outputs.

**Classification-oriented Metrics.** Classification tasks commonly adopt a series of standard metrics, including Accuracy, AUC, Equal Error Rate (EER), F1 Score, and Average Precision (AP/mAP). Accuracy reflects the proportion of correctly classified instances and is widely used in balanced datasets. AUC measures the discriminative ability of a model across varying thresholds and is particularly robust under class-imbalanced conditions. EER identifies the threshold at which false acceptance and false rejection rates are equal, which is especially relevant in authentication and identity verification scenarios. The F1 Score combines precision and recall into a single metric and applies to both image-level classification and pixel-level forgery detection. AP evaluates the precision-recall trade-off for individual classes, while mAP generalizes this evaluation across multiple classes or samples, making it well-suited for video forgery detection and temporal segment localization.

**Localization-oriented Metrics.** In localization tasks, evaluation metrics aim to assess the spatial or temporal precision of identifying manipulated regions. These metrics can be categorized as follows:

- Image-level Metrics. Common metrics include:

  - Intersection over Union (IoU): Measures the overlap between predicted and ground truth masks; widely used as a core indicator of localization accuracy.
  - Pixel-wise F1 Score: Captures both precision and recall at the pixel level; suitable for binarized probability maps.

- Pixel-wise Binary Classification Accuracy (PBCA): Computes overall pixel-wise accuracy, though it may be biased by the dominant non-manipulated class.
- Dice Score: Evaluates region-level overlap; often used both as an evaluation metric and a loss function in training.

- Text-level Metrics. In multimodal forgery detection, particularly for image-text pairs, manipulation may target the textual modality. Token-wise F1 Score evaluates a model's ability to identify which specific tokens or segments within a text sequence have been manipulated, functioning as a textual analogue of pixel-wise localization. In the context of our taxonomy, this metric directly supports the evaluation of Forgery Localization (FL) applied to the text modality: just as IoU measures how accurately a model localizes tampered image regions, Token-wise F1 measures how precisely manipulated textual segments are identified. Methods such as HAMMER (Shao et al., 2023), which detect cross-modal mismatches between images and accompanying text, rely on this metric to quantify the granularity of text-level tampering detection. This metric complements image-level localization metrics and is particularly relevant for multimodal scenarios where forgeries span both visual and textual content.

- Video-level Metrics. Metrics such as those used in DiMoDif (Koutlis & Papadopoulos, 2024) include:

  - AP@$p$: Measures average precision under an IoU threshold $p$ (*e.g.,* $0.5, 0.75, 0.95$).
  - AR@$n$: Computes average recall given up to $n$ candidate proposals; reflects the model's ability to retrieve manipulated segments over time.

- Multi-source Clustering Metrics. Some models such as SAFIRE (Kwon et al., 2025) use:

  - Adjusted Rand Index (ARI): Evaluates consistency between predicted and ground truth manipulation clusters. ARI is defined as:
  $$\text{ARI} = \frac{\text{RI} - \mathbb{E}[\text{RI}]}{\max(\text{RI}) - \mathbb{E}[\text{RI}]} \tag{5}$$

  where RI denotes the raw Rand Index. ARI corrects for chance and provides a normalized measure of agreement, making it suitable for segmentation consistency evaluation.

**Textual Explanation-oriented Metrics.** Assessing the quality of generated textual explanations in forgery detection involves a wide array of evaluation strategies, each emphasizing different aspects of language quality, semantic fidelity, or task utility. We summarize these methods into 5 categories:

- Text Similarity Metrics. These metrics measure surface-level lexical and syntactic similarity between generated explanations and reference texts.

  - N-gram Overlap (*e.g.,* BLEU-3/4) (Chakraborty et al., 2025; Guo et al., 2025d; Lian et al., 2024): BLEU evaluates the precision of contiguous word sequences (n-grams). BLEU-4 captures four-word phrase-level coherence. Higher scores indicate greater overlap in wording and phrasing.
  - Sequence Matching (*e.g.,* ROUGE-L) (Liu et al., 2024c; Kang et al., 2025): ROUGE-L computes the longest common subsequence between candidate and reference texts, capturing structural similarity.
  - Comprehensive Metrics (*e.g.,* CIDEr, METEOR) (Kundu et al., 2025a; Guo et al., 2025d): CIDEr incorporates TF-IDF weights to emphasize important words, while METEOR integrates synonym and stem matching. These metrics provide more nuanced assessments than n-gram precision alone, especially in multimodal settings.

- Semantic Consistency Metrics. These metrics assess deeper meaning alignment beyond lexical overlap.

  - Cosine Semantic Similarity (CSS) (Xu et al., 2024; Kang et al., 2025): Explanations are embedded using pretrained models (*e.g.,* MiniLM, BERT), and cosine similarity is computed between vectors. This accounts for paraphrasing and semantic equivalence.
  - Image–Text Consistency: This metric evaluates whether textual explanations align with visual evidence. ESIDE (Wu et al., 2025) employs such evaluation to validate that explanations describe manipulated content in the associated image.

- LLM-based Evaluation Frameworks. Large language models (LLMs) are increasingly adopted to evaluate explanations by simulating human judgment. Typically, the explanation is presented to the model together with contextual inputs such as the manipulated image or a reference response, formulated through a task-specific prompt. The LLM is then prompted to assign structured scores or output binary decisions.

  - Structured Scoring (*e.g.,* ForgeryAnalysis-Eval): FORGERYSLEUTH (Sun et al., 2024) uses GPT-4 to score explanations on a 1–10 scale across three axes: (i) object recognition accuracy, (ii) evidence relevance, and (iii) logical reasoning. This enables structured and reproducible evaluation.
  - Binary LLM-as-a-Judge (Kundu et al., 2025a; Yu et al., 2025b): Models like Gemini or GPT-4 output Yes/No decisions indicating whether the explanation agrees with the ground truth.

- Human Evaluation Protocols. Human raters provide subjective but trusted assessments of explanation quality.

  - Expert/Crowdsourced Ratings: $\chi^2$-DFD (Chen et al., 2024b) collects human feedback on clarity, correctness, and relevance using Likert scales or ranking. These evaluations offer fine-grained judgment often used as ground truth in benchmarks.
  - Task Utility Assessment (Chakraborty et al., 2025; Kwon et al., 2025; Huang et al., 2024a): User studies test whether explanations help improve task performance (*e.g.,* fake image identification), measuring practical effectiveness and explainability.

- Functional Assessment. These methods indirectly evaluate explanations through their impact on downstream tasks.

  - Performance Improvement: Systems like TRUTHLENS (Chakraborty et al., 2025) and SHIELD (Shi et al., 2024) report increased classification accuracy when explanations are incorporated, implying functional value.
  - Cross-Modal Consistency: FFAA (Huang et al., 2024a) ensures that visual evidence (*e.g.,* heatmaps) and textual explanations refer to the same manipulated content, validating coherence across modalities.

**Mapping Metrics to Explainability Dimensions.** To connect the evaluation metrics discussed above with the taxonomy proposed in this survey, we briefly summarize their alignment with FL, FA, and FJB. Forgery Localization is directly assessed by localization-oriented metrics such as IoU, pixel-wise F1, Dice Score, and Token-wise F1, which measure the spatial or textual precision of identified manipulated regions. It is also partially captured by text similarity metrics (*e.g.,* BLEU, ROUGE-L) when the generated explanation includes spatial descriptions of tampered areas. Forgery Attribution is primarily evaluated through classification-oriented metrics such as Accuracy, F1 Score, and AUC applied to multi-class attribution labels, and it can also be assessed through LLM-based and human evaluation protocols when attribution reasoning is part of the generated explanation. Forgery Judgment Basis, as the most open-ended dimension, relies most heavily on explanation-oriented metrics: text similarity and semantic consistency metrics evaluate the linguistic quality and faithfulness of generated rationales, LLM-based frameworks assess logical coherence and evidence relevance, human evaluation protocols provide trusted judgments on explanation quality, and functional assessments measure whether explanations improve downstream task performance. This mapping highlights that while FL and FA benefit from well-established quantitative metrics, FJB evaluation remains comparatively underdeveloped and largely dependent on reference-based text metrics or subjective human judgment—an observation that reinforces the need for standardized explainability benchmarks discussed in next section.

# 7 Challenges and Future Directions

With the rapid advancement of AI-generated content technologies in recent years, the interplay between generation and detection has increasingly resembled a two-player game, akin to a cat-and-mouse dynamic (Lin et al., 2024). As generation methods evolve, they continuously expose the limitations of current detection techniques. In response, the field faces a number of open challenges that remain insufficiently addressed. To advance the robustness and transparency of forgery detection, we highlight several promising research directions spanning methodology, dataset curation, and evaluation protocols.

**Explainable Detection Outputs.** One major challenge in forgery detection lies in the absence of structured and standardized formats for explanation generation. Although some existing models offer limited explainability, most rely on ad hoc techniques such as free-form textual rationales or saliency maps that highlight suspicious regions. Although these forms provide partial insight, they often fall short of delivering comprehensive and consistent justifications for the decisions of the model. Critically, few systems offer explanations that address all key dimensions of forensic analysis: (1) accurate localization of manipulated regions, (2) source attribution, such as identifying the generative model or the type of manipulation, and (3) explicit judgment basis that outlines the rationale of the decision. The lack of such multi-dimensional outputs significantly limits explainability and accountability. Establishing these three explanation dimensions as standard outputs could form a foundation for transparent, auditable, and trustworthy detection systems. Notable exceptions like FAKESHIELD (Xu et al., 2024; Huang et al., 2024a) demonstrate the feasibility of this approach by producing tripartite explanations that jointly cover localization, manipulation type, and decision rationale. However, such designs remain rare. Future research should therefore explore methods to guide model outputs toward these richer explanation formats. Promising directions include task-specific instruction tuning and explanation-aware multi-task learning objectives, which explicitly condition model outputs on locality, attribution, and reasoning targets. Additionally, modular architectures that decouple detection from explanation could enable more dedicated explainability modeling. Finally, incorporating structured output formats inspired by recent advances in vision-language reasoning tasks would promote consistency and human-readability across systems. Advancing these directions would enable the development of explanation-rich detectors that not only identify manipulations but also articulate how and why a given decision is made.

**Uncertainty-Aware Detection.** Existing methods largely prioritize accuracy, with limited attention to predictive confidence. In high-risk applications such as misinformation detection and legal forensics, the absence of uncertainty estimation can lead to overreliance on unreliable outputs. While a small subset of recent works has begun to explore uncertainty modeling to improve system robustness (Guillaro et al., 2023; Kwon et al., 2025; Huang et al., 2024a; Chakraborty et al., 2025), many of these efforts remain limited in scope. Some methods, such as TRUFOR (Guillaro et al., 2023) and SAFIRE (Kwon et al., 2025), provide explicit or weakly explicit confidence estimates for localized predictions. However, others rely on implicit indicators of uncertainty. For example, FFAA (Huang et al., 2024a) employs a Multi-answer Intelligent Decision System (MIDS) to evaluate the consistency of responses generated under different hypotheses, using it as a proxy for sample difficulty. TRUTHLENS (Chakraborty et al., 2025) includes textual labels such as *"confidence: high"* in its explanations, but these lack grounding in formal probabilistic estimation. Established techniques such as Bayesian neural networks, deep ensembles, and temperature scaling offer principled foundations for calibrated uncertainty estimation. Future work should investigate how these methods can be effectively integrated into forgery detection pipelines, enabling models to output explainable and trustworthy confidence measures and improving their reliability in real-world applications.

**Data Coverage and Annotation Gaps.** Forgery detection remains constrained by the limited scope, diversity, and annotation granularity of current datasets. Although several curated datasets have facilitated recent progress, many remain confined to narrow domains (*e.g.,* facial imagery, social media content, or synthetic benchmarks). As a result, they often fail to represent emerging manipulation techniques, including both traditional editing methods and content generated by modern generative AI models. Many datasets include visual-text pairs, but only a small subset involve human verification. This lack of verification makes it difficult to assess label reliability, particularly for fine-grained manipulations. Furthermore, most datasets provide only binary authenticity labels (real or fake) and lack detailed annotations such as manipulation type, affected image regions, or source-target attribution. These limitations hinder the development of robust, generalizable, and explainable forgery detection systems. To address these gaps, future datasets should be designed to cover a broader range of manipulation types and encompass more modalities and languages. They should also provide richer annotations such as region-level masks, semantic consistency judgments, manipulation provenance, and human-authored rationales. Such enhancements would better support multi-task learning and facilitate attribution-focused research.

**Real-World Applicability and Deployment Challenges.** An important but still underexplored challenge is real-world applicability. Existing benchmarks have been invaluable for measuring progress, yet they

only partially approximate the conditions under which forgery detection systems are actually deployed. In real environments, manipulated content emerges from diverse and continuously changing sources, passes through multiple platform-dependent compression and editing pipelines, and is often embedded in broader social, temporal, or multimodal contexts that are absent from curated datasets. This creates a fundamental distribution gap between benchmark evaluation and deployment. For explainable forgery detection, the challenge is even more pronounced: in many practical settings, the system is not expected merely to output a real/fake label, but to provide evidence that can be inspected, contested, and acted upon by humans. Such evidence may include localized traces of manipulation, attribution to a likely editing or generative pipeline, confidence or uncertainty estimates, and natural-language rationales that help users understand the basis and limitations of the model's judgment. Consequently, real-world utility depends not only on detection accuracy, but also on whether explanations remain stable under domain shift, meaningful across heterogeneous content types, and trustworthy enough to support high-stakes applications such as forensic investigation, news verification, regulatory compliance, and content moderation. Addressing this gap will require more realistic benchmarks, stronger evaluation under open-world conditions, and explainability frameworks that are designed for decision support rather than only post hoc illustration.

**Benchmark for Explainable Forgery Detection.** Recent advances in forgery detection have introduced a variety of explanation modalities, including visual heatmaps for manipulation localization, source attribution of the forgery (*e.g.,* identifying the source generative model), and textual rationales that articulate the basis for detection decisions. Despite these developments, the field lacks a unified framework or standardized metrics for assessing the quality, faithfulness, and consistency of such outputs. This gap complicates meaningful comparison between methods and presents a barrier to reproducibility, as current evaluation practices remain largely qualitative and task-specific. This motivates the need for systematic evaluation protocols and benchmark datasets that can support reliable, cross-method assessment of explainability in both unimodal and multimodal detection scenarios. This challenge is amplified by evaluation heterogeneity across localization paradigms. As shown in Section 4.1 and Table 2, even pixel-level mask methods are difficult to compare fairly due to differences in training data and evaluation protocols. The fragmentation is more severe for text-based explanation methods, where no consensus on evaluation methodology exists: different methods adopt LLM-based scoring, cosine semantic similarity, n-gram overlap metrics such as BLEU and ROUGE-L, or human Likert-scale ratings, each capturing fundamentally different aspects of explanation quality (lexical overlap, semantic alignment, and reasoning correctness). At the paradigm level, pixel-level methods use spatial metrics (IoU, F1), frame-level methods use temporal metrics (*e.g.*, AP@$p$, AR@$n$), and text-based methods use the heterogeneous metrics described above, making cross-method and cross-paradigm comparison largely infeasible. Future work should develop unified frameworks that jointly assess spatial precision, semantic correctness, and explanation quality within a single protocol, enabling meaningful comparison across localization paradigms.

**Limited Explainability in Video-based Detection.** Explainability for video forgery detection remains significantly underdeveloped. While many detectors operate on temporal data to predict real or fake labels, only a few methods explicitly address explainability, and among them, only two provide explainable forgery localization outputs. Most existing approaches are confined to static frame-level visualizations, offering limited insight into motion inconsistencies or temporally distributed manipulation patterns. Structured outputs such as source attribution or judgment rationales grounded in temporal evidence are even more scarce. This gap is particularly limiting given that many forgeries exploit subtle cross-frame artifacts, which static explanations fail to capture. Future research should explore techniques such as temporal attention visualization, prototype-based motion analysis, and video-language models capable of generating temporally grounded natural language explanations. Defining structured explanation formats for video, encompassing when a manipulation occurs, where it appears, and why it is detected, will be essential for developing explainable and trustworthy video forgery detection systems.

# 8   Broader Impact and Ethical Considerations.

As a survey of forgery detection methods, this work primarily synthesizes and organizes existing publicly available research rather than introducing new detection or generation techniques. Nevertheless, we recognize several ethical dimensions that warrant discussion.

First, by systematically cataloguing detection strategies, feature vulnerabilities, and architectural designs, this survey could in principle inform adversarial efforts to develop forgeries that evade the reviewed detection mechanisms. This tension between open scientific disclosure and potential misuse is inherent to security-related research. We believe that the benefits of transparent, reproducible research outweigh the risks, as the detection community relies on shared knowledge to advance the field collectively. Restricting access to detection methodologies would disproportionately hinder defenders while offering limited protection against well-resourced adversaries who can independently discover these techniques.

Second, we caution against over-reliance on the notion of explainability in forgery detection. While the taxonomy proposed in this survey (FL, FA, FJB) provides a structured framework for explainable outputs, current methods vary considerably in the depth and reliability of their explanations. In particular, some manipulations may leave no perceptible artifacts, making faithful explanation inherently difficult or even ill-defined. Deploying systems that claim to provide explanations without adequate grounding could lead to misplaced trust, especially in high-stakes contexts such as legal proceedings or journalistic verification. Practitioners should critically evaluate the quality and limitations of model-generated explanations rather than treating them as authoritative forensic evidence.

Third, forgery detection systems carry the risk of disparate impact across demographic groups. Models trained predominantly on specific populations or cultural contexts may exhibit uneven performance, potentially leading to biased outcomes in content moderation or forensic applications. Future work should prioritize fairness-aware evaluation and diverse dataset construction to mitigate these risks.

## 9 Conclusion

In summary, this survey has synthesized recent progress in explainable visual forgery detection and traced the field's evolution from binary classification to multi-dimensional explainability. We demonstrated the complementary nature of four feature-driven strategies (RGB, frequency, noise-texture, and representation learning) and examined how single-modal and multi-modal methods approach localization, attribution, and reasoning tasks. Despite these advances, significant challenges remain, including the absence of standardized explanation protocols, limited uncertainty quantification, insufficient dataset diversity, and underexplored video-based explainability. Addressing these issues requires principled frameworks, robust benchmarks, and methods capable of handling temporal consistency. Looking forward, we argue that forgery detection systems must achieve not only high accuracy but also transparency and accountability to earn trust in real-world deployments. The proposed taxonomy and identified research directions provide a roadmap for developing reliable and explainable detection technologies that can serve critical applications in forensic analysis, journalism, and governance.

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

## 10 Appendix

### 10.1 Survey Methodology

To ensure transparency and allow readers to assess the completeness and potential bias of this survey, we describe the literature search and selection protocol followed.

**Databases and Sources.** We conducted our literature search primarily through Google Scholar and arXiv. Google Scholar was used as the main discovery tool due to its broad coverage of peer-reviewed venues, preprints, and cross-references. arXiv was monitored continuously to capture recent preprints not yet published in conference or journal proceedings. In addition, we performed backward and forward citation tracking on key papers identified during the initial search to uncover related works that may not have appeared directly in keyword-based queries.

**Search Terms.** We used combinations of the following core keywords and their variants: "forgery detection," "fake detection," "deepfake detection," "forgery localization," "manipulation explanation," and "AI-generated image detection." These terms were combined with modifiers such as "explainable," "survey," "foundation model," "large language model," and "vision-language model" to capture the growing intersection of generative AI and explainable detection.

**Time Range.** Our search primarily targeted works published from 2023 onward, reflecting the rapid emergence of explainable forgery detection as a distinct research direction. Earlier foundational works (*e.g.*, traditional forgery detection methods and seminal generative models) were included where necessary to provide essential context, but the core focus of the survey is on methods proposed within the past three years.

**Inclusion Criteria.** A method was included if it satisfied the following conditions:

- The work addresses visual forgery detection (including video and multimodal settings where images are the primary modality).

- The method provides at least one form of explainable output beyond binary real/fake classification, such as manipulation localization (masks, bounding boxes, or textual descriptions), source attribution, or decision reasoning.

- The work was published at a top-tier venue (*e.g.*, CVPR, ICCV, ECCV, NeurIPS, ICML, ICLR, AAAI, IJCAI, ACM MM, ACM CCS) or appeared as a publicly available preprint with significant community attention (*e.g.*, citation count, code availability, or relevance to the explainability focus of this survey).

**Exclusion Criteria.** We excluded works that:

- Focus exclusively on binary classification without any explainability component.

- Address forgery detection in modalities other than images (*e.g.*, pure text or pure audio forgery detection) without an visual component.

- Are duplicates or earlier versions of papers that were later published in an extended form (in such cases, the most recent version was included).

**Selection Process.** The initial keyword search yielded a broad set of candidate papers. These were screened in two stages: first by title and abstract to assess relevance to explainable forgery detection, and then by full-text reading to verify that the inclusion criteria were met. The final set comprises 48 methods, which we believe represents the current landscape of explainable forgery detection. We acknowledge that this is a rapidly evolving field, and some concurrent or very recent works may not be captured. We have made our best effort to include all relevant methods available at the time of writing, and we welcome the community to bring any omissions to our attention.

