# OpenReview forum: "Explainable Visual Forgery Detection: A Survey"
_TMLR — Rejected by TMLR_

### Review · Reviewer_BCir · 2026-03-03

**Summary Of Contributions:**

The paper presents a survey of recent work on the problem of forgery detection under the lens of explainability. The authors provide a high-level categorization of research into several related problems: forgery localization, forgery attribution and forgery judgement basis and review 48 methods, by also looking into feature-driven strategies, datasets and benchmarking strategies and discussing some limitations and challenges.

Strengths:
- The survey is well written and quite compact, which makes it easy to read and get an overview of recent research.
- The survey addresses an important aspect of the problem, namely explainability (even though not really comprehensively).
- The provided taxonomy and tables with references are quite useful as a structured and compact means of presenting the surveyed methods.

Weaknesses:
- While the focus of the survey is Explainability in forgery detection, a large part of the paper is actually dedicated to the classic problem formulations of detection, localization and attribution, which have been thoroughly reviewed by previous works.
- There is no explanation of the survey methodology followed. It appears as if the authors picked a relatively small sample of well-known methods in the area and presented them under the devised taxonomy.
- The survey lacks in-depth coverage of performance aspects, and it is not clear whether the three problems FL, FA, FJB are solved and to what degree.
- There are certain presentation issues that could be improved (cf. Requested Changes).

**Audience:**

Yes

**Audience Explanation:**

The survey is a useful and interesting read for people working in the area. It is quite high-level and avoids excessive technical detail, so it should be more appealing to people with little or moderate experience and knowledge in the area. Having said that, the percentage of TMLR audience who would be interested in this work is likely quite small as the problem is focused (despite gaining research interest over the last few years).

**Broader Impact Concerns:**

The paper does not contain an impact statement. Such a statement could be quite useful, especially with respect to a) discussing the challenge of open disclosure versus the risks involved in developing adversarial methods that can evade detection (of which the development is facilitated through open disclosure); b) discussing caveats and risks of "promising" explainability even in cases when this is not feasible or is ill-defined (e.g. non-visible artifacts),

**Claims And Evidence:**

No

**Claims Explanation:**

The paper claims to provide a systematic review of the field and to lay the foundations of future transparent and trustworthy forgery
detection systems, supporting real-world applications in forensic analysis, news verification, and regulatory compliance. These claims are in my view not properly supported by the presented work.

- The review is not really systematic. There is no description of how the presented 48 methods were selected (queries, search engines, inclusion/exclusion criteria, etc.), which makes it look as if the survey cherry-picked familiar methods while leaving out potentially interesting and relevant methods. Also, for such a large and growing field of study, 48 methods seems rather limited.

- While the focus of the survey is explainability, the survey dedicates a large part of the text to repeating/representing the well-known problem formulations that previous surveys have largely covered. I understand that there is a trade-off between making the survey self-contained and making it focused, but my feeling was that explainability was not sufficiently explored.

- The authors loosely refer to both explainability and interpretability throughout the text, while there are differences between the two. This could create confusion.

- While the devised taxonomy and structure are reasonable and helpful, I feel that within each subsection, the authors simply summarize the reviewed methods using 2-3 lines per work. The tables are helpful as an overview of the main method attributes; however, I would expect some more in-depth insights and understanding of the methods, including their strengths and limitations. For instance, the authors mention some basic information like the underlying architecture used per method (even though not in a very standard way), but do not discuss aspects such as computational requirements, training setup and limitations.

- Evaluation of explainability is briefly touched during the later part of the paper. The authors acknowledge that this is a relatively under-studied aspect of the problem. However, I would still feel that the paper could expand further on the topic, offering more details and perhaps some visual aids to the different approaches used to assess explainability.

**Requested Changes:**

- The survey should expand considerably the number of reviewed methods and provide the protocol used to collect the methods. Even if less methods are presented in more detail, it is important to get a much broader view of the area and acknowledge relevant works and developments across the globe. The same also applies for the reviewed datasets (with a quick glance I noticed some important well-known datasets and benchmarks missing, e.g. SynthBuster, SIDBench, TGIF).

- The authors should be more precise about the main concepts used, e.g. explainability vs interpretability, and introduce them early on to avoid ambiguities and improve clarity.

- The authors should reconsider the title of the paper, as the survey is not only about image forgeries (as it covers partly audio and video forgery detection), and also reconsider whether explainability is the key unique contribution of the paper (and therefore featured in the title) or not.

- The authors should better justify the distinction of the different task settings, e.g. why do they distinguish between Photoshop-generated and AI-generated forgeries.

- The survey should provide a more in-depth and insightful treatment of the reviewed methods (cf. comments above) and also present more quantitative data about their performance as a means of capturing the extent to which the presented tasks/problems are currently solved.

- The survey should also discuss more extensively the real-world applicability of methods in the literature by discussing aspects such as tools (cf. verification plugin), user interfaces, generalization to in-the-wild cases, robustness to adversarial attacks, etc.

- The authors should address the ethical and impact aspects of the underlying research (cf. Broader Impact Concerns).

- The diagrams could be refined, e.g. by selecting a more neutral font (e.g. calibri, arial, raleway) compared to comic sans.

---

> ### Author Response · Authors · 2026-03-27
> **Author Response and Revised Version Summary**
>
> We sincerely thank the reviewer for the detailed and constructive feedback. We have revised the manuscript to address the requested changes point by point.
>
> >**[Requested Change 1] Expand the reviewed methods/datasets and provide the protocol used to collect them**
>
> In the revised manuscript, we now explicitly provide the survey methodology by pointing readers to **Appendix 10.1**, where we describe the literature search strategy, including the databases, search terms, time range, and inclusion/exclusion criteria. We also expanded the dataset coverage in **Table 5**; in particular, **TGIF**, **SynthBuster**, and **SIDBench** are now included.
>
> >**[Requested Change 2] Be more precise about the main concepts, especially explainability vs. interpretability**
>
> We thank the reviewer for highlighting this important conceptual distinction. In the revised manuscript, we clarify that our survey is centered on **explainability**, rather than interpretability, and we have unified the terminology and overall framing accordingly.
>
> >**[Requested Change 3] Reconsider the title and scope of the paper**
>
> We revised the title accordingly. The manuscript is now titled **“Explainable Visual Forgery Detection: A Survey”**. We also clarified the scope in the first paragraph of the **Introduction**, where we explicitly state that visual forgery detection covers not only images, but also video and image-centric multimodal content.
>
> >**[Requested Change 4] Better justify the distinction between different task settings, such as Photoshop-style and AI-generated forgeries**
>
> Our intention is not merely to distinguish two sources of fake content, but to justify why they correspond to different forgery attribution settings. In our framework, **Forgery Attribution (FA)** explains *how* a forgery was produced. Photoshop-generated and AI-generated forgeries therefore correspond to different attribution targets: the former involves traditional editing operations, while the latter involves learned generative models. Because they differ substantially in manipulation pipelines, forensic traces, and attribution granularity, we treat them as different task settings rather than a single homogeneous category.
>
> This rationale is reflected in three places: (1) the **Introduction**, which describes the evolution from traditional editing to modern AI-driven generation; (2) **Section 3.1**, which is divided into Traditional Image Forgery Methods and AI-Driven Image Forgery Methods; and (3) the **Figure 2** attribution taxonomy, which includes “AIGC vs. Traditional Editing” as one attribution level.
>
> >**[Requested Change 5] Provide a more in-depth and quantitative treatment of the reviewed methods**
>
> We appreciate the reviewer’s suggestion. To address it, we introduced revisions aimed at moving beyond method-by-method enumeration. At the same time, fully standardized comparison remains challenging because existing methods often differ substantially in evaluation protocols, benchmark datasets, task formulations, and reporting settings. Even when two methods study forgery localization, they may focus on different manipulation types, use different dataset splits, or report different metrics. Nevertheless, to provide readers with a more consistent view, we added **Table 2** and a comparative analysis paragraph at the end of **Section 4.1**, where we synthesize broader patterns in localization performance and explicitly note the limitations of fair comparison under non-unified protocols.
>
> >**[Requested Change 6] Discuss real-world applicability more extensively**
>
> We appreciate this suggestion. In the revised manuscript, we added a dedicated discussion of **real-world applicability and deployment challenges** in **Section 7**. This new paragraph emphasizes the gap between curated benchmark evaluation and real-world deployment, where manipulated content is more diverse, open-ended, and platform-dependent, and where binary predictions are often insufficient without localization, attribution, uncertainty estimates, or human-readable rationales.
>
> >**[Requested Change 7] Address ethical and broader-impact aspects**
>
> In response, we added a new **Section 8, “Broader Impact and Ethical Considerations.”** This section explicitly discusses: (1) the tension between open disclosure and the risk of helping adversaries evade detection, (2) the danger of over-promising explainability in settings where explanations may be unreliable or ill-defined, and (3) the possibility of disparate impact across demographic groups.
>
> >**[Requested Change 8] Improve diagram presentation**
>
> We appreciate this presentation suggestion. We revised all figure fonts to Calibri.

---

### Review · Reviewer_UK1C · 2026-03-15

**Summary Of Contributions:**

The paper presents a survey into the problem of explainable image forgery. The authors suggest the need of changing the perspective of forgery detection, from 'whether' to 'where', 'how', and 'why'.  They structure the survey according to the following taxonomy of methods:

(1) forgery localisation (FL) (answering which part of an image has been affected by a forgery)

(2) forgery attribition (FA) (answering by what means the manipulation has been performed)

(3) forgery judgement basis (FJB) (answering why the system has decided that it may be a forgery)

**Audience:**

Yes

**Audience Explanation:**

I think this work has interest to the community, because there is a need to systematise explainable image forgery, and there is an audience of researchers and policymakers who are interested in image forgery detection.

**Broader Impact Concerns:**

No broader impact concerns as it summarises the existing openly-accessible methods and evaluation protocols

**Claims And Evidence:**

Yes

**Claims Explanation:**

The claims are supported by the evidence as the authors claim and then present a comprehensive review of the techniques behind the aforementioned taxonomy. There are, however, a number of points which I think needs to be addressed

**Requested Changes:**

However, there are also some concerns about structure, that I think need to be addressed:
- it seems to me that while up to Section 4, the paper sticks to the topic, which is Explainable Forgery detection. In Section 5, however, it swerves into generic forgery detection strategies, without particular focus on the topic of the review. I would think connecting it better with the rest of the paper would be highly beneficial. In this case, would it be possible to make it clear how these methods are a part of explainable forgery detection as opposed to some generic one?
- As a linked but separate point, it seems like Section 5 loses the link of the methods with the particular identified types  of forgery detection (FL, FA, FJB). I think that explaining how different methods serve the purpose of every of these problems would help unify the narrative.
- In Section 6.4, Explanation-Oriented Metrics, I would suggest answering the questions how these metrics address different types of forgery explanation (FL, FA, FJB).

Finally, which is a minor point, I would check the paper for the typos (e.g. Page 22 should read: 'Textual Explanation-oriented Metrics', Page 8 should have whitespaces before the brackets, e.g. 'Forgery Localization (Section 4.1)').

---

> ### Author Response · Authors · 2026-03-27
> **Author Response and Revised Version Summary**
>
> We sincerely thank the reviewer for these constructive suggestions. Below we respond to the three structural concerns point by point.
>
> >**[Requested Change 1] Section 5 should be more clearly connected to explainable forgery detection**
>
> We appreciate this important observation. In response, we revised the opening paragraph of **Section 5** to make the connection to explainable forgery detection explicit. Rather than presenting feature extraction as a generic design choice, the revised text now frames Section 5 from an **explainability perspective**, asking what kinds of explainable outputs each feature paradigm can provide and what limitations it entails. This change was intended to better connect Section 5 with the overall theme of the survey and with the explainability taxonomy introduced earlier.
>
> >**[Requested Change 2] Section 5 should better link methods to FL, FA, and FJB**
>
> We thank the reviewer for this suggestion. To address it, we added **Section 5.5, "The Role of Features in Explainability,"** together with **Table 4**. This new part explicitly maps the four feature types (**RGB, Frequency, Noiseprint, Representation**) to the three explainability dimensions (**FL, FA, FJB**). The implications of this mapping are then discussed systematically through **Observations 1--4** in **Section 5.5**. This revision was intended to make clearer how different feature-driven methods serve different explainability purposes, rather than treating them as generic forgery detection strategies.
>
> >**[Requested Change 3] Section 6.4 should explain how metrics address FL, FA, and FJB**
>
> We appreciate this valuable comment. In the revised manuscript, we added a dedicated discussion in the end of Section 6.4, **"Mapping Metrics to Explainability Dimensions,"** which explicitly connects major metric families to **FL, FA, and FJB**. This revision was intended to make the metric discussion more consistent with the taxonomy and to clarify how different evaluation protocols correspond to different explainability objectives.

---

> > ### Comment · Reviewer_UK1C · 2026-04-01
> >
> > Dear Authors,
> >
> > Many thanks for your thorough rebuttal. I have read the whole set of other authors' responses and the rebuttal itself, however there's one additional aspect. In the interest of facilitating reading the updated version and to fully confirm how the new revision addresses the comments, could you please provide a version with marked changes?
> >
> > Many thanks,
> > the Reviewer

---

> ### Author Response · Authors · 2026-04-01
> **Author Response and Revised Version Summary**
>
> Thank you very much for your careful follow-up and for pointing this out. We appreciate the importance of making the revision easier to inspect.
>
> In response, we have prepared and uploaded a marked-changes version of the revised manuscript, in which all added or modified text is highlighted in red. Please note that these marked changes reflect revisions made in response to the comments from all reviewers. Details of revisions based on the comments given by all reviewers are provided in **"Revision Summary (1/3), (2/3), and (3/3)"** above.
>
> With respect to your specific comments, the main revisions are located in the following parts of the manuscript:
>
> 1. **Opening paragraph of Section 5**
>    We revised the opening paragraph (in Page 14) to make the connection to explainable forgery detection more explicit.
>
> 2. **New Section 5.5 The Role of Features in Explainability and Table 4**
>    We added a new subsection (from page 19 to 20) and table that explicitly map the four feature types (RGB, Frequency, Noiseprint, Representation) to the three explainability dimensions (FL, FA, and FJB). We also added Observations 1–4 to summarize the main implications of this mapping.
>
> 3. **End of Section 6.4 Mapping Metrics to Explainability Dimensions**
>    We added a dedicated discussion at the end of Section 6.4 (in page 25) to explicitly connect major evaluation metrics to FL, FA, and FJB, so as to clarify how different metric families correspond to different explainability objectives.
>
> We hope the marked-changes version makes it easier to verify how the revised manuscript addresses your comments as well as those of the other reviewers.

---

> > ### Comment · Reviewer_UK1C · 2026-04-09
> >
> > I've checked the whole set of reviews and the revision, I would like to highlight that the changes helped improve the draft in the following way:
> > - Formalisation of the inclusion criteria in the Appendix alleviates some of the comments around the inclusion.
> > - comparative analysis is strengthened in a way that it summarises instead of retelling
> > - the link between the methods and the identified types of forgery detection (FL, FA, FJB) is now presented  in Section 5
> > - Section 5  is linked with the topic of explainability
> > - Unifying the terminology throughout the text (such as explainability), edits of the text throughout to the end of improving consistency
> >
> > Overall, the value of this work is that we need systematisation of existing methods for explainable visual forgery detection, given the number of papers available and the overall interest in the topic.
> >
> > I have the following question concerning the clarity of the following changes:
> > - the discussion of the methods around Table 2 looks unclear to me, e.g. : "Comparative Analysis of Localization Methods. Table 2 reports pixel-level F1 scores of representative localization methods across seven benchmark datasets. Three key patterns emerge. First, dual-stream methods that fuse noise-based and RGB features achieve consistently strong performance on traditional manipulation datasets." In which way does it follow from Table 2, which lists the names of the methods (vertically) and the metrics (horizontally)?

---

> > > ### Author Response · Authors · 2026-04-10
> > > **Author Response and Revised Version Summary**
> > >
> > > We sincerely appreciate the reviewer's feedback, it's very helpful in improving the clarity of our paper.
> > >
> > > In fact, Table 3 includes a "Feature-Driven Detection" column that categorizes the feature strategy of each method. However, we totally agree with the reviewer's observation: as a quantitative comparison table, Table 2 lacked this explicit link, making it difficult for readers to directly trace the patterns discussed in the text back to the corresponding methods. We thank the reviewer for pointing out this oversight.
> > >
> > > Following this suggestion, we have added a "Feature-Driven Detection" column to Table 2, consistent with the Feature-Driven Detection column in Table 3. This allows readers to directly verify the patterns summarized in the text that *“dual-stream methods fusing Noiseprint and RGB features achieve stronger performance on traditional manipulation benchmarks”*. For the reviewer's convenience, we attach the updated Table 2 below.
> > >
> > > ## Updated Table 2
> > >
> > > | Method | Feature-Driven Detection | CASIAv1+ | Coverage | Columbia | NIST16 | IMD20 | DSO-1 | CocoGlide |
> > > |--------|--------------------------|----------|----------|----------|--------|-------|-------|-----------|
> > > | TruFor | Noiseprint, RGB | 0.822 | 0.735 | 0.914 | 0.470 | -- | **0.973** | 0.720 |
> > > | ERMPC | Noiseprint, RGB | **0.876** | **0.944** | **0.968** | **0.895** | **0.856** | -- | -- |
> > > | EITLNet | Noiseprint, RGB | 0.530 | 0.448 | 0.881 | 0.308 | 0.532 | -- | 0.410 |
> > > | CSR-Net | Rep. | 0.585 | 0.780 | -- | 0.835 | -- | -- | -- |
> > > | Zhu et al. | Noiseprint, RGB | 0.621 | 0.812 | -- | 0.868 | -- | -- | -- |
> > > | DA-HFNet | Noiseprint, RGB, Freq. | -- | -- | -- | -- | -- | -- | 0.585 |
> > > | AdaIFL | Rep., RGB | 0.848 | 0.745 | -- | 0.706 | -- | -- | -- |
> > > | ForgeryGPT | Rep. | 0.569 | -- | 0.773 | 0.549 | 0.530 | 0.506 | -- |
> > > | Forma | Freq. | 0.729 | 0.587 | 0.949 | 0.454 | -- | -- | 0.453 |
> > > | GIFL | Rep. | 0.783 | 0.565 | -- | -- | -- | -- | 0.571 |
> > > | FakeShield | Rep. | 0.600 | -- | 0.750 | 0.370 | 0.570 | 0.520 | -- |
> > > | ForgerySleuth | Rep. | 0.870 | 0.792 | 0.931 | 0.610 | -- | -- | **0.751** |
> > > | EditScout | Rep. | -- | -- | -- | -- | -- | -- | 0.457 |
> > >
> > >
> > > We will update the revised paper accordingly. We sincerely thank the reviewer again for the constructive feedback throughout the review process.

---

> ### Comment · Reviewer_UK1C · 2026-04-14
>
> Many thanks for the updated version, I think it addresses the discrepancy between Table 2 and its interpretation in the text.

---

### Review · Reviewer_YWHv · 2026-03-15

**Summary Of Contributions:**

The authors present a comprehensive survey on the field of explainable image-centric forgery detection. The paper transitions the focus of forgery detection from binary "real vs. fake" classification to a structured interpretability framework. Key contributions include:
1. Proposing a novel taxonomy based on three dimensions: Forgery Localization (FL), Forgery Attribution (FA), and Forgery Judgment Basis (FJB)
2. Systematically analyzing 48 state-of-the-art methods across single-modal and multi-modal settings.
3. Reviewing four feature-driven detection strategies: RGB, frequency-domain, noise-texture, and representation learning.
4. Synthesizing benchmark datasets, evaluation protocols, and open challenges for the field.

**Strengths**:
1. The taxonomy is highly intuitive and addresses a critical need for transparency in high-stakes domains like forensic analysis and news verification.
2. The inclusion of multi-modal generative manipulation is timely and well-structured.

**Weaknesses**:
1. The manuscript occasionally lists methods rather than critically synthesizing their trade-offs. For example, Section 4.1 simply enumerates pixel-level mask methods like TRUFOR, MUN, HIFI-NET, MSCCNET, and AEROBLADE, without comparing their relative computational costs or localization precision.
2. Section 5 shifts toward generic forgery detection features rather than maintaining a strict focus on how these features specifically enable explainability.
3. The discussion of evaluation metrics in Section 6.4 is superficial and disjointed from the taxonomy. For instance, the "Text-level Metrics" subsection consists of a single sentence regarding the "Token-wise F1 Score", completely failing to explain how this metric evaluates Forgery Localization (FL).
4. Proposed future directions in Section 7 are often too generic. The "Uncertainty-Aware Detection" subsection broadly lists techniques like "Bayesian neural networks, deep ensembles, and temperature scaling" without exploring how these probabilistic estimates could be integrated into structured Forgery Judgment Basis (FJB) explanations.
5. The manuscript contains formatting and typographical errors that detract from its quality. Examples include citation errors like "Yn et al., (Yu et al., 20248)" in Table 1 and typos such as "singel-modal" in Section 5 and "Texual" in Section 6.4.

**Additional Comments:**

Visual aids in the paper are excellent. Figure 1 provides a highly effective roadmap of the field's rapid evolution, and Figure 3 successfully demonstrates the complex concepts of FL, FA, and FJB with digestible visual examples.

Strengthening the critical analysis will elevate this paper from a good summary to an essential reference for the community.

**Audience:**

Yes

**Audience Explanation:**

Yes, the paper is of interest to the TMLR community that includes researchers on Compute Vision, Generative AI, and Multimedia Generation, and can be explained form the perspectives listed below:
* The rapid advancement of large-scale diffusion models like Stable Diffusion, FLUX, and Midjourney has made traditional binary detection techniques sometimes inadequate (Wang et al.)
* There is a high demand for systems that are transparent, auditable, and human-explainable. (Guo et al.)
* By structuring the scattered literature into a cohesive taxonomy of "where" (FL), "how" (FA), and "why" (FJB), this survey provides a highly valuable roadmap for researchers building the next generation of trustworthy AI systems

Wang, Z., Bao, J., Zhou, W., Wang, W., Hu, H., Chen, D., & Li, H. (2023). "DIRE for Diffusion-Generated Image Detection." Proceedings of the IEEE/CVF International Conference on Computer Vision (ICCV), pp. 22445-22455.
Guo, Z., et al. (2025). "Rethinking Vision-Language Model in Face Forensics: Multi-Modal Interpretable Forged Face Detector." Proceedings of the IEEE/CVF Conference on Computer Vision and Pattern Recognition (CVPR).

**Broader Impact Concerns:**

The survey inadvertently provides a blueprint for malicious actors to refine generative models to bypass these exact detection mechanisms, and thus a brief Broader Impact statement should be added to acknowledge this.

**Claims And Evidence:**

Yes

**Claims Explanation:**

The authors successfully substantiate their proposed taxonomy by mapping existing literature directly to FL, FA, and FJB categories.
* They provide extensive evidence through detailed comparison tables, such as Table 1 mapping related surveys and Table 2 detailing architecture and feature strategies.
* The progression of the field toward deeper explainability is effectively supported by tracking the timeline of representative approaches like TRUFOR and FAKESHIELD from 2023 onward.
* However, the claim of providing a "systematic analysis" is occasionally hindered by a lack of comparative critique: the authors describe what each method does but rarely evaluate their relative computational complexities or failure scenarios.

**Requested Changes:**

1. Synthesize rather than enumerate methods (Section 4.1): The current text lists pixel-level mask methods (e.g., TRUFOR, MUN, HIFI-NET, MSCCNET, and AEROBLADE) but lacks a comparative critique. A discussion synthesizing their specific trade-offs, such as computational overhead, failure modes, and relative localization precision, would be welcome to see.

2. Refocus feature extraction on explainability (Section 5): This section currently reads as a generic summary of forgery detection features (RGB, Frequency, Noiseprint, Representation). It's better connect each feature strategy back to the proposed taxonomy, detailing exactly how a specific feature type enables FL, FA, and FJB.

3. Expand evaluation metrics (Section 6.4): The discussion of metrics is superficial and disjointed from the core taxonomy. For example, the "Text-level Metrics" subsection consists of only a single sentence ("Token-wise F1 Score is used to evaluate whether the model accurately identifies manipulated segments within a text sequence"). Please expand this to explain exactly how this metric evaluates FL or FJB outputs.

4. Correct typographical and formatting errors
Table 1: Fix citation formatting errors, such as "Yn et al., (Yu et al., 20248)" and "Zou et al, Zou et al., 2025)".
Section 5: Correct the spelling of "singel-modal".
Section 6.4: Correct the heading typo "Texual Explanation-oriented Metrics", also in paragraph.

---

> ### Author Response · Authors · 2026-03-27
> **Author Response and Revised Version Summary**
>
> We sincerely thank the reviewer for the thoughtful and insightful feedback. Below are our responses.
>
> >**[Weakness 1 / RC1] Synthesize rather than enumerate methods (Section 4.1)**
>
> We appreciate the reviewer’s suggestion to strengthen the comparative discussion in Section 4.1. Due to the diversity of model architectures, training settings, benchmark choices, and evaluation protocols, establishing a fully standardized comparison in this area remains challenging. In particular, even when methods address the same forgery localization task, they are often evaluated on different datasets, under different splits, or with different reporting settings, which limits strict one-to-one comparison. Nevertheless, to provide readers with a more consistent and informative view, we made our best effort in the revised manuscript to synthesize the available evidence across representative methods. Specifically, we added **Table 2** and a corresponding comparative analysis paragraph at the end of **Section 4.1**, where we summarize three broader patterns in localization performance and discuss their implications. We also added a **cross-category analysis** in the same part of **Section 4.1** to explain why the other FL paradigms currently lack directly comparable benchmarks and metrics. In addition, the limitation of fair comparison due to the lack of a unified protocol is now explicitly noted there and further discussed in **Section 7**.
>
> >**[Weakness 2 / RC2] Refocus feature extraction on explainability (Section 5)**
>
> We are grateful for this sharp observation, which helped us improve the focus of Section 5. In response, we made two revisions. First, we **rewrote the opening paragraph of Section 5** to frame feature extraction from an explainability perspective, asking what kinds of explainable outputs each feature paradigm can provide and what limitations it entails. Second, we added **Section 5.5, "The Role of Features in Explainability,"** together with **Table 4**, which maps the four feature types (RGB, Frequency, Noiseprint, Representation) to the three explainability dimensions (**FL, FA, FJB**). The detailed implications of this mapping are discussed in **Section 5.5** through **Observations 1--4** and summarized in **Table 4**.
>
> >**[Weakness 3 / RC3] Expand evaluation metrics (Section 6.4)**
>
> We thank the reviewer for pointing out this gap. In the revision, we strengthened **Section 6.4** in two ways. First, we expanded **Text-level Metrics** and now explain explicitly that **Token-wise F1** is the **textual analogue of pixel-wise localization**, thus directly evaluating **FL in the text modality**. Second, we added **"Mapping Metrics to Explainability Dimensions,"** which explicitly connects major metric families to **FL, FA, and FJB**. This change better aligns the metric discussion with our taxonomy and also clarifies that **FJB evaluation remains comparatively underdeveloped**.
>
> >**[Weakness 4] Uncertainty-Aware Detection (Section 7) is too generic**
>
> We appreciate this valuable suggestion. We agree that systematic consideration of uncertainty remains limited in the current literature. Existing works mainly reflect uncertainty through output-level signals, such as confidence maps, cross-response consistency, or textual confidence indicators, rather than through principled uncertainty modeling that is tightly integrated with explainable forensic reasoning. We therefore clarify in **Section 7** that uncertainty-aware detection is still underexplored, particularly for **FJB-style explanations**, and that future work should develop more systematic and calibrated approaches to uncertainty estimation.
>
> >**[Weakness 5 / RC4] Correct typographical and formatting errors**
>
> We thank the reviewer for the careful reading and attention to detail. We corrected the issues identified by the reviewer in the revised manuscript, including: **Table 1** citation formatting; **Section 5** ("singel-modal" → "single-modal"); **Section 6.4** ("Texual" → "Textual"); missing whitespace before parenthetical section references in **Section 4**; and additional proofreading throughout.
>
> >**[Broader Impact Concerns] Broader Impact statement**
>
> We appreciate the reviewer's suggestion to make the broader implications more explicit. Accordingly, we added **Section 8, "Broader Impact and Ethical Considerations,"** which discusses: (1) the tension between open disclosure and adversarial misuse, (2) the risk of over-reliance on explainability in high-stakes settings, and (3) the possibility of disparate impact across demographic groups.
>
> We are grateful again for the reviewer's constructive and forward-looking comments.

---

### Author Response · Authors · 2026-04-01
**Revision Summary (1/3)**

Based on all the valuable comments and suggestions, we have revised the manuscript accordingly (highlighted in red). Below, we provide a structured summary of the main revisions in the hope that it will make the changes easier to review.

**1. Revised the title (Page 1, Title) [Reviewer BCir - RC3]**

This revision addresses the issue that the original title did not fully reflect the actual scope of the paper. We changed the title to “Explainable Visual Forgery Detection: A Survey” by removing “Image-Centric” so that the title more accurately covers images, videos, and image-centric multimodal forgery detection.

**2. Added a scope definition sentence (Page 1, Introduction, first paragraph) [Reviewer BCir - RC3]**

This revision addresses the issue that the scope of the paper was not sufficiently explicit. We added a sentence in the first paragraph of the Introduction to clarify that “visual forgery detection” in this survey refers primarily to image forgery detection, while also covering video and image-centric multimodal content.

**3. Added a survey methodology reference sentence (Page 3, end of the Introduction) [Reviewer BCir - RC1]**

We added a sentence after the contribution list to direct readers to Appendix 10.1, where the literature search strategy, databases, search terms, time range, and inclusion/exclusion criteria are described in detail.

**4. Added Table 2: performance comparison of Forgery Localization methods (Page 11, Section 4.1) [Reviewer YWHv - RC1; Reviewer BCir - RC5]**

We added Table 2 to summarize the pixel-level F1 scores of representative localization methods on seven benchmark datasets, providing a more direct basis for performance comparison.

**5. Added a “Comparative Analysis of Localization Methods” paragraph (Pages 11–12, end of Section 4.1) [Reviewer YWHv - RC1; Reviewer BCir - RC5]**

We added a comparative analysis paragraph after Table 2 to summarize several key patterns, including the strong performance of dual-stream methods, the large performance variation across datasets, and the trade-off between semantic explainability and spatial precision in multi-modal approaches. We also explicitly note the limitations of fair comparison under non-unified evaluation protocols.

**6. Added a “Cross-Category Analysis of FL Approaches” paragraph (Page 12, end of Section 4.1) [Reviewer YWHv - RC1]**

We added a cross-category analysis paragraph to explain why instance-level, bounding-box, frame-level, and text-based FL approaches are not yet directly comparable to pixel-level mask methods, and clarified that they rely on different evaluation systems such as spatial IoU/F1, temporal AP@p, and NLP-based metrics.

**7. Rewrote the opening paragraph of Section 5 (Page 14, Section 5) [Reviewer YWHv - RC2; Reviewer UK1C - RC1/RC2]**

We completely rewrote the opening paragraph of Section 5 so that feature extraction is introduced from an explainability perspective, focusing on what kinds of explainable outputs different feature paradigms can support and what limitations they have.

**8. Added Section 5.5 “The Role of Features in Explainability” and Table 4 (Pages 19–20) [Reviewer YWHv - RC2; Reviewer UK1C - RC2]**

We added a new subsection and Table 4 to systematically map the four feature types (RGB, Frequency, Noiseprint, and Representation) to the three explainability dimensions FL, FA, and FJB. We also added Observations 1–4 to summarize the main roles and limitations of these feature types in explainability.

**9. Added TGIF, SynthBuster, and SIDBench to Table 5 (Page 21, Table 5) [Reviewer BCir - RC1]**

We added TGIF, SynthBuster, and SIDBench to the public-source dataset portion of Table 5 in order to make the dataset review more complete.

**10. Rewrote the opening paragraph of Section 6.4 (Page 23, Section 6.4) [Reviewer YWHv - RC3; Reviewer UK1C - RC3]**

We rewrote the opening paragraph to organize the metrics into three groups (classification-oriented, localization-oriented, and explanation-oriented), and clarified how each group corresponds to FL, FA, and FJB.

---

### Author Response · Authors · 2026-04-01
**Revision Summary (2/3)**

**11. Expanded the discussion of Text-level Metrics (Page 24, Section 6.4) [Reviewer YWHv - RC3]**

We expanded the original one-sentence subsection into a full paragraph, explicitly explaining that Token-wise F1 is the textual analogue of pixel-wise localization and therefore directly evaluates FL in the text modality. We also clarified its role in multimodal forgery scenarios.

**12. Added a “Mapping Metrics to Explainability Dimensions” paragraph (Page 25, end of Section 6.4) [Reviewer YWHv - RC3; Reviewer UK1C - RC3]**

We added a new paragraph at the end of Section 6.4 to systematically connect major metric families to FL, FA, and FJB, and to clarify that FJB evaluation remains comparatively underdeveloped.

**13. Added a “Real-World Applicability and Deployment Challenges” paragraph (Pages 26–27, Section 7) [Reviewer BCir - RC6]**

We added a new paragraph in Section 7 discussing the gap between curated benchmark evaluation and real-world deployment, including open-ended content diversity, platform dependence, and the higher requirements for explanation reliability and usability in practice.

**14. Revised “Benchmark for Explainable Forgery Detection” paragraph (Page 27, Section 7) [Reviewer YWHv - RC1; Reviewer BCir - RC5]**

We added a new paragraph in Section 7 to discuss evaluation heterogeneity across localization paradigms, pointing out that even pixel-level mask methods are difficult to compare fairly because of differences in training data, dataset splits, and reporting protocols, while text-based methods suffer from even more fragmented evaluation settings.

**15. Added Section 8 “Broader Impact and Ethical Considerations” (Pages 27–28) [Reviewer YWHv - Broader Impact; Reviewer BCir - RC7]**

We added a complete new section discussing three aspects: the tension between open disclosure and adversarial misuse, the risk of over-promising explainability when explanations are unreliable or ill-defined, and the possibility of disparate impact across demographic groups.

**16. Added Appendix 10.1 “Survey Methodology” (Page 42) [Reviewer BCir - RC1]**

We added a full appendix describing the survey methodology, including databases and sources, search terms, time range, inclusion criteria, exclusion criteria, and the overall selection process.

---

### Author Response · Authors · 2026-04-01
**Revision Summary (3/3)**

Below are some formatting issues in the paper that we corrected:

**1. Unified the use of “explainability” and “interpretability” terminology (throughout the manuscript) [Reviewer BCir - RC2]**

We revised the manuscript to consistently use “explainability” as the central term and corrected expressions that could cause conceptual ambiguity, so that the overall framing is clearer and more consistent.

**2. Corrected typographical and formatting errors and conducted additional proofreading (multiple locations) [Reviewer YWHv - RC4; Reviewer UK1C - RC4]**

We corrected multiple issues, including citation formatting errors in Table 1, the spelling correction from “singel-modal” to “single-modal” in Section 5, the correction from “Texual” to “Textual” in Section 6.4, and missing spaces before parenthetical section references. We also conducted additional proofreading throughout the manuscript.

**3. Revised all figure fonts to Calibri (all figures) [Reviewer BCir - RC8]**

We replaced the figure fonts with Calibri throughout the manuscript so that the visual presentation is more consistent, neutral, and professional.



**We again sincerely thank all the reviewers for their valuable comments and constructive suggestions. We believe these revisions have substantially strengthened the manuscript, and we hope that the above summary helps make the changes easier to review.**

---

### Decision · Action_Editor_wPFg · 2026-05-21

**Recommendation:** Reject

**Additional Comments:**

N/A

**Audience:**

Yes

**Audience Explanation:**

The topic is of interest to the community.

**Claims And Evidence:**

No

**Claims Explanation:**

While the authors have addressed most issues in the revised paper, the reviewers still express substantial concerns. The main issues are reproducibility and scope (comprehensivity), as rightly pointed out by one expert reviewer. The claims made in this submission are not clearly supported in this paper due to these reasons. These are critical issues that need to be adequately addressed before they may be considered for publication. Since there is no major revision option, the only possible decision is to reject this work in its current form.

**Resubmission Of Major Revision:**

The authors may consider submitting a major revision at a later time.